# PLAN-R1: SAFE AND FEASIBLE TRAJECTORY PLANNING AS LANGUAGE MODELING

**Xiaolong Tang, Meina Kan, Shiguang Shan, Xilin Chen**
Institute of Computing Technology, Chinese Academy of Sciences
University of Chinese Academy of Sciences
{tangxiaolong22s, kanmeina, sgshan, xlchen}@ict.ac.cn

## ABSTRACT

Safe and feasible trajectory planning is critical for real-world autonomous driving systems. However, existing learning-based planners rely heavily on expert demonstrations, which not only lack explicit safety awareness but also risk inheriting undesirable behaviors such as speeding from suboptimal human driving data. Inspired by the success of large language models, we propose Plan-R1, a two-stage trajectory planning framework that decouples principle alignment from behavior learning. In the first stage, a general trajectory predictor is pre-trained on expert data to capture diverse, human-like driving behaviors. In the second stage, the model is fine-tuned with rule-based rewards using Group Relative Policy Optimization (GRPO), explicitly aligning ego planning with principles such as safety, comfort, and traffic rule compliance. This two-stage paradigm retains human-like behaviors while enhancing safety awareness and discarding undesirable patterns from demonstrations. Furthermore, we identify a key limitation of directly applying GRPO to planning: group-wise normalization erases cross-group scale differences, causing rare, high-variance safety-violation groups to have similar advantages as abundant low-variance safe groups, thereby suppressing optimization for safety-critical objectives. To address this, we propose Variance-Decoupled GRPO (VD-GRPO), which replaces normalization with centering and fixed scaling to preserve absolute reward magnitudes, ensuring that safety-critical objectives remain dominant throughout training. Experiments on the nuPlan benchmark demonstrate that Plan-R1 significantly improves planning safety and feasibility, achieving state-of-the-art performance, particularly in realistic reactive settings. Our code is available at https://github.com/XiaolongTang23/Plan-R1.

## 1 INTRODUCTION

Trajectory planning is a fundamental component of autonomous driving systems, directly influencing vehicle safety, efficiency, and the ability to navigate complex and dynamic environments. In recent years, learning-based planning approaches (Hu et al., 2023; Xu et al., 2024) have attracted increasing attention due to their strong adaptability, competitive performance, and minimal reliance on manually designed rules. These methods offer promising solutions for generating trajectories that can respond effectively to various traffic scenarios and rapidly changing road conditions.

However, due to the complexity and diversity of real-world driving scenarios, most existing planning methods, whether based on imitation learning (IL) (Cheng et al., 2024b;a; Zheng et al., 2025; Tang et al., 2025) or reinforcement learning (RL) (Zhang et al., 2025; Hu et al., 2024; Li et al., 2025), rely heavily on expert demonstrations for supervision. This dependency introduces two key limitations: (i) expert data rarely covers negative scenarios such as collisions or off-road driving, leaving the model unable to explicitly learn how to avoid them, and (ii) demonstrations are not always optimal and may contain undesirable behaviors. For example, we find that over 10% of the nuPlan (Caesar et al., 2021) training scenes exhibit speeding, and some contain uncomfortable maneuvers or critically low time-to-collision (TTC) values. As a result, models trained purely on expert data risk inheriting these undesirable behaviors without a clear notion of safety, as shown in Figure 2.

To address these limitations, we draw inspiration from the success of large language models (LLMs) (Shao et al., 2024; Ouyang et al., 2022), which typically adopt a two-stage training paradigm: pre-training as a general-purpose predictor via next-token prediction, followed by RL fine-tuning to align outputs with desired objectives (e.g., format). **Our key insight is that trajectory planning can be formulated in a similar way, i.e., first trained as a trajectory predictor on expert driving data, and then fine-tuned using RL to align the trajectory with explicit planning principles, such as safety, comfort, and rule compliance.** This decoupled paradigm separates principle alignment from behavior learning, enabling the model to retain human-like behaviors, enhancing safety awareness and discarding undesirable patterns learned from the expert data.

Specifically, we introduce Plan-R1, a two-stage dual-model framework for principle-aligned trajectory planning. In the pre-training stage, trajectories are discretized into motion tokens across time and space (Seff et al., 2023; Philion et al., 2023), and a general motion predictor is trained with next-motion-token prediction to capture diverse, human-like multi-agent behaviors. In the fine-tuning stage, unlike prior methods that rely on human preference data (Huang et al., 2024) or additional expert demonstrations (Li et al., 2025), which may introduce biases or undesirable behaviors, we use rule-based rewards that provide consistent and unbiased supervision. To ensure realistic multi-agent interactions and stable optimization, we introduce a dual-model design: a trainable ego planner explores alternative decisions, while a frozen copy of the pre-trained model serves as a reactive world model to predict the responses of surrounding agents. This separation enables ego-centric policy updates without destabilizing non-ego behaviors, yielding stable, interaction-aware joint predictions.

For RL optimization, we adopt Group Relative Policy Optimization (GRPO) (Shao et al., 2024), which has shown strong performance across various domains (Huang et al., 2025). However, we find that its default design is not well suited for trajectory planning. Unlike mathematical reasoning tasks where the goal is simply to produce the correct answer, trajectory planning must jointly optimize multiple, potentially conflicting objectives with carefully designed priorities. For example, collision avoidance must always take precedence over comfort. After pre-training, nearly 80% of trajectory groups contain no safety violations, and their reward variance is dominated by non-safety objectives such as comfort. Standard GRPO normalizes rewards independently within each group, erasing natural scale differences across groups. **This causes rare, high-variance safety-violation groups to have normalized advantages comparable to abundant safe groups (Figure 4), diluting safety-critical gradients and shifting optimization toward non-safety objectives.** To address this, we propose Variance-Decoupled GRPO (VD-GRPO), which replaces per-group normalization with centering and fixed scaling. By preserving absolute reward magnitudes, safety-critical groups naturally generate larger gradients, ensuring that high-priority safety objectives remain dominant and continue to improve even in late-stage training when such cases become extremely rare.

By addressing both behavior-principle decoupling and long-tailed safety optimization, Plan-R1 provides a unified and scalable solution for safe and feasible trajectory planning. Results on the nuPlan benchmark show that Plan-R1 significantly improves both safety and feasibility, achieving state-of-the-art performance and demonstrating strong generalization to challenging interactive scenarios.

The primary contributions of this paper are:

- We introduce a new perspective that formulates trajectory planning as a principle-aligned prediction task, decoupling planning principle alignment from behavior learning to overcome the limitations of expert data.

- We propose Plan-R1, a two-stage dual-model framework that first pre-trains a motion predictor on expert data to capture diverse driving behaviors, and then fine-tunes it with rule-based reinforcement learning, requiring no additional expert or preference data.

- We identify a key limitation of GRPO: per-group normalization erases cross-group scale differences, diluting safety-critical signals. We address it with Variance-Decoupled GRPO (VD-GRPO) to preserve absolute reward magnitudes, prioritizing safety-critical objectives during optimization.

- Plan-R1 achieves state-of-the-art performance on the nuPlan benchmark, significantly improving both safety and feasibility, particularly in challenging reactive settings.

## 2 RELATED WORK

### 2.1 LEARNING-BASED TRAJECTORY PLANNING

Learning-based trajectory planning methods can be broadly grouped into imitation learning (IL), reinforcement learning (RL), and hybrid IL+RL approaches. Pure IL and RL have been extensively explored for trajectory planning and achieved notable progress (Bansal et al., 2018; Scheel et al., 2022; Tang et al., 2025; Cheng et al., 2024b;a; Zhang et al., 2025; Li et al., 2024; Cusumano-Towner et al., 2025). However, they both have inherent limitations: IL relies heavily on expert demonstrations, which rarely cover safety-critical events and often contain suboptimal behaviors such as speeding; RL enables behavior discovery beyond demonstrations but suffers from severe sample inefficiency, complex multi-objective reward design, and poor human-likeness in large, dynamic environments.

Recent works therefore explore hybrid IL+RL approaches. One line of research adopts imitation-regularized RL, where the distance to expert trajectory is incorporated into the optimization objective as either a reward signal or a regularizer (Zhang et al., 2025; Lu et al., 2023). For example, BC-SAC (Lu et al., 2023) jointly optimizes a behavior cloning loss and a Soft Actor-Critic loss to improve safety. These methods stabilize training but remain heavily dependent on expert data, inheriting its biases and undesirable behaviors. Another line follows a pre-training + fine-tuning paradigm inspired by LLMs, such as Gen-Drive (Huang et al., 2024) and TrajHF (Li et al., 2025). Gen-Drive trains a reward model from collected preference data and fine-tunes the planner via RL. TrajHF fine-tunes the planner using human driving preference data to align with desired driving styles. However, collecting preference data is expensive and may introduce new biases, limiting scalability and reliability. In contrast, our Plan-R1 also adopts a two-stage framework but replaces preference data with rule-based rewards, which provide consistent and unbiased supervision. This allows us to align planning with safety and other principles while preserving the diverse human-like behaviors, achieving safe and robust trajectory planning without extra human supervision.

### 2.2 GROUP RELATIVE POLICY OPTIMIZATION (GRPO)

GRPO (Shao et al., 2024) is a reinforcement learning algorithm originally developed for mathematical reasoning. It samples multiple rollouts under the same context to form a group and normalizes rewards within each group to compute relative advantages, eliminating the need for a value function and avoiding instability from inaccurate value estimation in methods like PPO (Schulman et al., 2017). This simplicity makes GRPO highly effective for alignment tasks such as reasoning (Huang et al., 2025) and search (Jin et al., 2025). Moreover, DAPO (Yu et al., 2025) extends GRPO by introducing asymmetric clipping and dynamic sampling to improve optimization stability and efficiency. Dr.GRPO (Liu et al., 2025) removes the per-group standard deviation term, aiming to reduce problem-level difficulty bias and stabilize optimization in language model training. VD-GRPO is inspired by Dr.GRPO but addresses a fundamentally different problem in a different domain: in safety-critical trajectory planning, rewards are multi-objective and often conflicting, where rare but catastrophic events (e.g., collisions) must be prioritized. We demonstrate that group-wise normalization erases cross-group scale differences, causing rare, high-variance safety-violation groups to have similar advantages to abundant low-variance safe groups, suppressing optimization for safety-critical objectives. VD-GRPO address it by replacing normalization with centering and fixed scaling, ensuring that safety-critical objectives remain dominant throughout training.

## 3 METHOD

We propose Plan-R1, a two-stage framework that decouples planning principle alignment from behavior learning. As illustrated in Figure 1, Plan-R1 consists of an autoregressive pre-training stage and a rule-based RL fine-tuning stage. The model is first pre-trained on expert demonstrations to capture diverse human-like motion behaviors, and then fine-tuned with rule-based rewards to align the ego motion with explicit planning principles. In subsection 3.1, we formulate trajectory planning as a principle-aligned sequence prediction task. subsection 3.2 describes the autoregressive pre-training process, while subsection 3.3 presents the fine-tuning stage and introduces Variance-Decoupled GRPO (VD-GRPO) to further address the safety-critical optimization challenge.

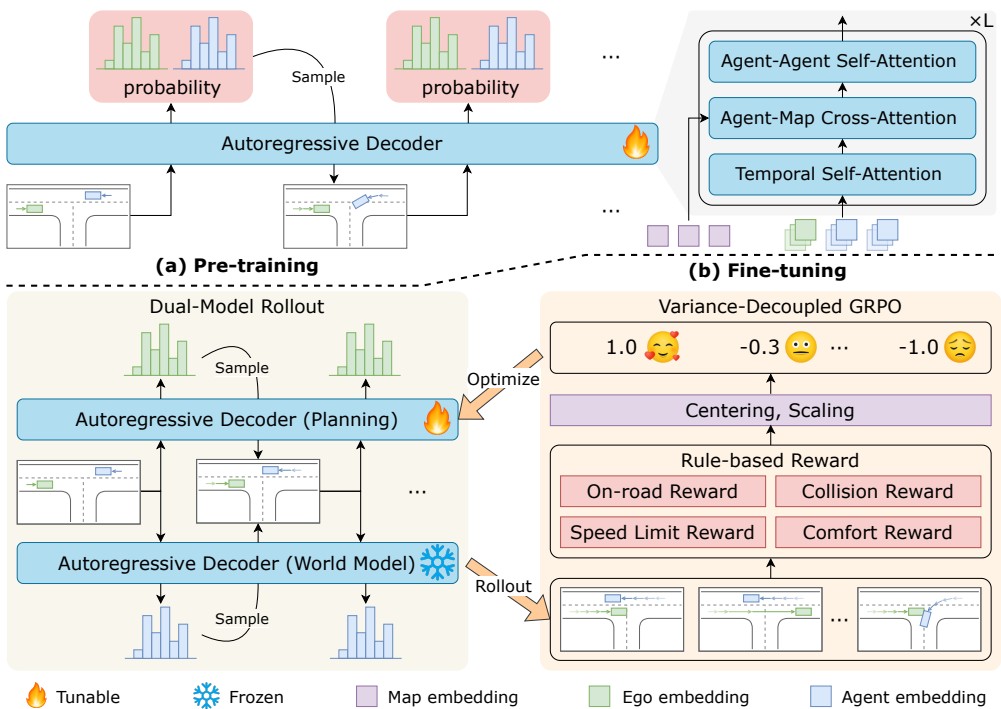

Figure 1: Illustration of our Plan-R1: Stage (a) pre-trains a motion predictor on expert data; Stage (b) fine-tunes it with VD-GRPO using rule-based rewards to align with planning principles.

## 3.1 PROBLEM FORMULATION

Trajectory planning is inherently sequential, making autoregressive modeling a natural choice. We extend this formulation into a principle-aligned autoregressive prediction task by explicitly incorporating high-level planning principles (e.g., safety, comfort, traffic rule compliance). In this formulation, the ego vehicle's trajectory is generated jointly with the predicted motions of surrounding agents (Hagedorn et al., 2024; Zheng et al., 2024a; Chen et al., 2024), enabling the planner to model realistic multi-agent interactions. Formally, given a scene context $C$ and a set of planning principles $P$, our goal is to generate the joint future motions $Y = \{y_{t,n}\}_{t=1:F, n=0:N}$ for all agents, where $F$ is the planning horizon, $N$ is the number of surrounding agents, $y_{t,n}$ denotes the motion of agent $n$ at time step $t$, and the ego vehicle is indexed by $n = 0$. Directly modeling $p(Y \mid C, P)$ is intractable, but leveraging temporal causality (Seff et al., 2023) and short-term conditional independence (Tang & Salakhutdinov, 2019; Rhinehart et al., 2019) allows factorization as:

$$
\begin{aligned}
p(Y \mid C, P) &= \prod_{t=1}^{F} p(y_{t,0:N} \mid y_{<t,0:N}, C, P) \\
&= \prod_{t=1}^{F} p(y_{t,0} \mid y_{<t,0:N}, C, P) \prod_{n=1}^{N} p(y_{t,n} \mid y_{<t,0:N}, C, P) \qquad (1)\\
&\approx \prod_{t=1}^{F} \underbrace{\pi_e(y_{t,0} \mid y_{<t,0:N}, C, P)}_{\text{ego planner}} \prod_{n=1}^{N} \underbrace{p_a(y_{t,n} \mid y_{<t,0:N}, C)}_{\text{agent predictor}},
\end{aligned}
$$

where the last approximation assumes that surrounding agents' future motions are independent of the ego vehicle's planning principles $P$. This yields two sub-problems: (i) predicting each surrounding agent's motion $p_a(y_{t,n} \mid y_{<t,0:N}, C)$, and (ii) generating the ego vehicle's trajectory $\pi_e(y_{t,0} \mid y_{<t,0:N}, C, P)$ with additional conditioning on planning principles. The agent predictor $p_a$ can be learned via next-motion prediction on large-scale datasets (Wu et al., 2024), whereas

modeling the ego planner $\pi_e$ is more challenging because it must not only produce human-like motions while satisfying $P$. Since $p_a$ is trained on large-scale human driving data that implicitly reflects $P$, it provides a strong prior for $\pi_e$. However, this alignment is only implicit and imperfect: human demonstrations may still contain unsafe or suboptimal behaviors, and $p_a$ itself cannot guarantee strict adherence to $P$. To address this, we initialize $\pi_e$ with $p_a$ and fine-tune it using reinforcement learning, where rule-based rewards explicitly ensure compliance with $P$.

## 3.2 AUTOREGRESSIVE PRE-TRAINING

**Tokenization.** To enable autoregressive modeling, continuous trajectories are first discretized into motion tokens. Temporally, trajectories are segmented at fixed intervals. Spatially, the K-disk clustering algorithm (Philion et al., 2023) is applied to the resulting motion segments based on their average corner distance, producing a motion token vocabulary. Each token represents a prototypical displacement and heading change over a fixed time step. For each agent, the trajectory is thus transformed into a sequence of motion tokens spanning the prediction horizon.

**Model architecture.** Our model employs a transformer decoder with factorized attention (Nayakanti et al., 2023; Ngiam et al., 2021) to model multi-agent spatio-temporal interactions, as shown in Figure 1 (a). The architecture is designed to capture the temporal dynamics of each agent's trajectory and its interactions with the map and neighboring agents. Following prior work (Wu et al., 2024; Seff et al., 2023), motion tokens are processed through a stack of attention-based fusion blocks. Each block consists of three components: Temporal self-attention, modeling motion continuity and capturing temporal dependencies for each agent; Agent-map cross-attention, integrating spatial constraints by attending to road elements and ensuring compliance with the road network; Agent-agent cross-attention, capturing local interactions with neighboring agents to reason about dynamic behaviors. To maintain translation and rotation invariance, relative spatio-temporal position embeddings (Zhou et al., 2023; Tang et al., 2024) are applied to all attention modules. Both ego and non-ego agents share the same encoder structure, and all attention computations are implemented in a query-centric manner for efficient multi-agent rollout.

**Training objective.** The model is trained with a next-motion-token prediction objective, minimizing the negative log-likelihood of the next token for all agents:

$$\mathcal{L}_{\text{pretrain}} = -\sum_{t=1}^{F} \sum_{n=0}^{N} \log p_a(y_{t,n} \mid y_{<t,0:N}, C). \tag{2}$$

This objective enables the model to approximate the latent distribution of human driving behaviors, capturing diverse and realistic motion patterns from large-scale expert data. At this stage, the model simply imitates human-like behavior without explicit planning principles.

## 3.3 REINFORCEMENT LEARNING FINE-TUNING

While autoregressive pre-training captures human-like motion patterns, it does not explicitly enforce high-level planning principles. To address this gap, we fine-tune the ego motion predictor using reinforcement learning, as shown in Figure 1 (b), optimizing it to better align with predefined planning principles such as safety, comfort, and rule compliance. The generation process is formulated as a sequential decision-making problem, where the ego agent iteratively selects motion tokens to maximize rewards that reflect desirable planning behaviors.

**Dual-Model rollout.** A key challenge in RL fine-tuning is to realistically simulate how surrounding agents react to the ego vehicle's actions. A naive solution is to replay ground-truth (GT) behaviors of surrounding agents, which ignores ego interventions and thus yields unrealistic, non-reactive simulations. We address this issue with a dual-model design: a trainable ego planner $\pi_e$ interacts with a frozen copy of the pre-trained model $p_a$, which acts as a reactive world model for surrounding agents. During rollouts, $\pi_e$ explores alternative decisions while $p_a$ predicts the responses of other agents based on the evolving joint history, enabling interaction-aware simulation without requiring additional training for $p_a$. This separation allows ego-centric policy updates without destabilizing non-ego dynamics, resulting in stable and realistic multi-agent rollouts.

**Rule-based Rewards.** Unlike pure RL-based planning, our approach does not need to learn realistic human-like behaviors from scratch. This is because the pre-trained model already generates plausible, human-like motions, which greatly simplifies reward design: it only needs to target specific aspects such as safety, comfort, and rule compliance, without the burden of modeling basic driving realism. Here, we design a set of interpretable, rule-based reward functions covering key aspects such as collision avoidance, driving area compliance, comfort, speed limit compliance, and progress. Following (Caesar et al., 2021; Hu et al., 2024), we compute the total reward as the product of two components: a set of multiplicative safety indicators (i.e., collision avoidance, driving area compliance) and a weighted sum of soft cost terms (e.g., comfort, speed limit compliance, progress):

$$R(y_t) = \prod_{k \in \mathcal{I}_{\text{safe}}} \mathbf{1}_{k,t} \cdot \sum_{j \in \mathcal{I}_{\text{cost}}} w_j \cdot r_j(y_t), \tag{3}$$

where $\mathbf{1}_{k,t} \in \{0, 1\}$ denotes whether safety constraint $k$ is satisfied at step $t$, and $r_j(y_t)$ is the score of cost term $j$ with weight $w_j$. This formulation ensures that violations of critical safety conditions will nullify the total reward, while soft planning objectives are optimized only when safety constraints are satisfied. Details of the reward implementation are provided in Appendix C.

**Variance-Decoupled GRPO.** We adopt Group Relative Policy Optimization (GRPO) (Shao et al., 2024) to fine-tune the ego policy. GRPO eliminates the need for an explicit value function by computing relative advantages within each sampled group, reducing implementation complexity and avoiding instability caused by inaccurate value estimation.

During fine-tuning, the pre-trained trajectory predictor serves as a fixed reference policy $\pi_{\text{ref}}$, while the ego policy $\pi_e$ is updated to better align with planning principles. For each scenario, a group of $G$ future trajectories $\{Y^1, \ldots, Y^G\}$ is sampled from the old ego policy $\pi_{e_{\text{old}}}$, and the loss is:

$$\mathcal{L}_{\text{finetune}} = -\frac{1}{GF} \sum_{g=1}^{G} \sum_{t=1}^{F} \left[ \frac{\pi_e(y_t^g \mid C, P, y_{<t}^g)}{\pi_{e_{\text{old}}}(y_t^g \mid C, P, y_{<t}^g)} \hat{A}_t^g - \beta \, D_{\text{KL}}[\pi_e \parallel \pi_{\text{ref}}] \right], \tag{4}$$

where $\hat{A}_t^g = \sum_{\tau=t}^{F} \tilde{R}(y_\tau^g)$ is the cumulative advantage of token $t$. Standard GRPO normalizes rewards within each group as $\tilde{R}(y_t^g) = (R(y_t^g) - \mu_R)/\sigma_R$, with $\mu_R, \sigma_R$ being the group mean and standard deviation. The KL divergence term regularizes the update, keeping the fine-tuned policy close to the reference policy and thereby retaining human-like behaviors learned during pre-training.

However, directly applying GRPO to trajectory planning yields limited improvements. We attribute this to a key limitation of GRPO in multi-objective planning: group-wise normalization erases cross-group scale differences (Figure 4). As a result, rare safety-violation groups are normalized to have similar advantages as the abundant safe groups. Since most safe groups only exhibit minor fluctuations in comfort or other secondary objectives, critical safety signals become indistinguishable during optimization. Over time, the optimizer gradually shifts its focus toward these abundant non-safety objectives, leading to severe under-optimization of rare but catastrophic safety cases.

To address this issue, we propose Variance-Decoupled GRPO (VD-GRPO), which replaces per-group normalization with centering and a fixed global scaling constant $c$:

$$\tilde{R}^{\text{VD}}(y_t^g) = \frac{(R(y_t^g) - \mu_R)}{c}. \tag{5}$$

By decoupling normalization from variance, VD-GRPO preserves absolute reward scales across groups, ensuring safety-critical objectives remain dominant over secondary goals. High-variance groups, typically corresponding to rare catastrophic cases, naturally produce larger gradients, amplifying safety signals without manual reweighting. This enables continuous improvement on rare but important safety cases even in late-stage training.

## 4 EXPERIMENTS

### 4.1 EXPERIMENTAL SETUP

**Datasets.** We evaluate our method on the nuPlan benchmark (Caesar et al., 2021), a large-scale platform for trajectory planning in autonomous driving. The dataset contains over 1,300 hours of

Table 1: Comparison with SOTAs on nuplan benchmark. The best result is in **bold** and the second best result is underlined. *: with rule-based post-processing. NR/R: non-reactive/reactive mode.

| Type | Planner | Val14 | | Test14-hard | | Test14-random | |
|---|---|---|---|---|---|---|---|
| | | NR | R | NR | R | NR | R |
| Expert | Log-Replay | 93.53 | 80.32 | 85.96 | 68.80 | 94.03 | 75.86 |
| Rule-based & Hybrid | IDM (Treiber et al., 2000) | 75.60 | 77.33 | 56.15 | 62.26 | 70.39 | 72.42 |
| | PDM-Closed* (Dauner et al., 2023) | 92.84 | 92.12 | 65.08 | 75.19 | 90.05 | 91.64 |
| | PDM-Hybrid* (Dauner et al., 2023) | 92.77 | 92.11 | 65.99 | 76.07 | 90.10 | 91.28 |
| | Gameformer* (Huang et al., 2023) | 79.94 | 79.78 | 68.70 | 67.05 | 83.88 | 82.05 |
| | PLUTO* (Cheng et al., 2024a) | 92.88 | 89.84 | **80.08** | 76.88 | 92.23 | 90.29 |
| | PlanAgent* (Zheng et al., 2024b) | 93.26 | 92.75 | 72.51 | 76.82 | - | - |
| | Diffusion Planner* (Zheng et al., 2025) | 94.26 | 92.90 | 78.87 | **82.00** | **94.80** | 91.75 |
| | Carplanner* (Zhang et al., 2025) | - | - | - | - | 94.07 | 91.10 |
| | Plan-R1* (Ours) | **94.72** | **93.54** | 78.46 | 81.70 | 94.64 | **93.71** |
| Learning-based | UrbanDriver (Scheel et al., 2022) | 68.57 | 64.11 | 50.40 | 49.95 | 51.83 | 67.15 |
| | PDM-Open (Dauner et al., 2023) | 53.53 | 54.24 | 33.51 | 35.83 | 52.81 | 57.23 |
| | PlanTF (Cheng et al., 2024b) | 84.27 | 76.95 | 69.70 | 61.61 | 85.62 | 79.58 |
| | PLUTO (Cheng et al., 2024a) | 88.89 | 78.11 | 70.03 | 59.74 | 89.90 | 78.62 |
| | Diffusion Planner (Zheng et al., 2025) | **89.87** | 82.80 | 75.99 | 69.22 | 89.19 | 82.93 |
| | Plan-R1 (Ours) | 88.98 | **87.69** | **77.45** | **77.20** | **91.23** | **90.04** |

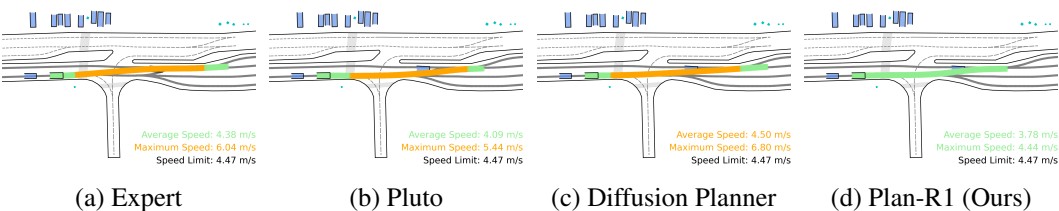

| (a) Expert | (b) Pluto | (c) Diffusion Planner | (d) Plan-R1 (Ours) |

Figure 2: Comparison of closed-loop ego trajectories. Trajectories are color-coded: orange indicates speeding segments, while green represents compliant motion. The expert trajectory (a) shows a clear speeding segment. Both PLUTO (b) and Diffusion Planner (c) mimic this behavior, indicating that planners trained solely on expert data tend to inherit undesirable patterns. In contrast, Plan-R1 (d) avoids speeding, demonstrating the effectiveness of rule-based reinforcement learning fine-tuning.

expert driving logs collected across four cities, covering diverse and challenging urban driving scenarios. Simulation runs for 15 seconds at 10 Hz: the ego vehicle executes its planned trajectory using a bicycle model and an LQR controller, while surrounding agents either follow replayed trajectories or react through the IDM (Treiber et al., 2000) policy. Following PLUTO (Cheng et al., 2024a), we sample 1M training instances for autoregressive pre-training. To reduce the computational cost of closed-loop rollouts during reinforcement learning, we build a smaller fine-tuning set of 100K scenarios. Evaluation is performed on the standard Val14, Test14-random, and Test14-hard splits (Dauner et al., 2023; Cheng et al., 2024b) under both non-reactive and reactive settings.

**Metrics.** We evaluate planning performance in closed-loop simulation using two standard metrics: Non-Reactive Closed-Loop Score (NR-CLS) and Reactive Closed-Loop Score (R-CLS). NR-CLS replays logged trajectories for surrounding agents, while R-CLS uses the IDM (Treiber et al., 2000) planner to generate reactive behaviors, providing a more challenging and realistic evaluation. Both metrics assess 15-second rollouts across key driving objectives such as collision avoidance and speed limit compliance, with scores ranging from 0 to 100 (higher is better).

## 4.2 COMPARISON WITH SOTAS

**Quantitative results.** We compare our method, Plan-R1, against a wide range of existing planners on the nuPlan benchmark, as shown in Table 1. Plan-R1 matches the strongest prior method, Diffusion Planner (Zheng et al., 2025), under the non-reactive setting, showing that RL fine-tuning

Table 2: Ablation study on the importance of ruled-based RL fine-tuning and VD-GRPO.

| Planner | NR-CLS | Collision | TTC | Drivable | Speed | Comfort | Progress | R-CLS |
|---|---|---|---|---|---|---|---|---|
| Pre-training only | 85.61 | 94.83 | 90.04 | 94.64 | 96.57 | 99.62 | 91.64 | 82.81 |
| + GRPO | 88.65 | 93.87 | 91.57 | 96.93 | **99.65** | 99.62 | **94.11** | 88.35 |
| + VD-GRPO (Ours) | **91.23** | **97.32** | **95.02** | **97.32** | 99.45 | **99.62** | 91.94 | **90.04** |
| Δ (Ours vs. Pre-training) | +5.62 | +2.49 | +4.98 | +2.64 | +2.88 | +0.00 | +0.30 | +7.23 |

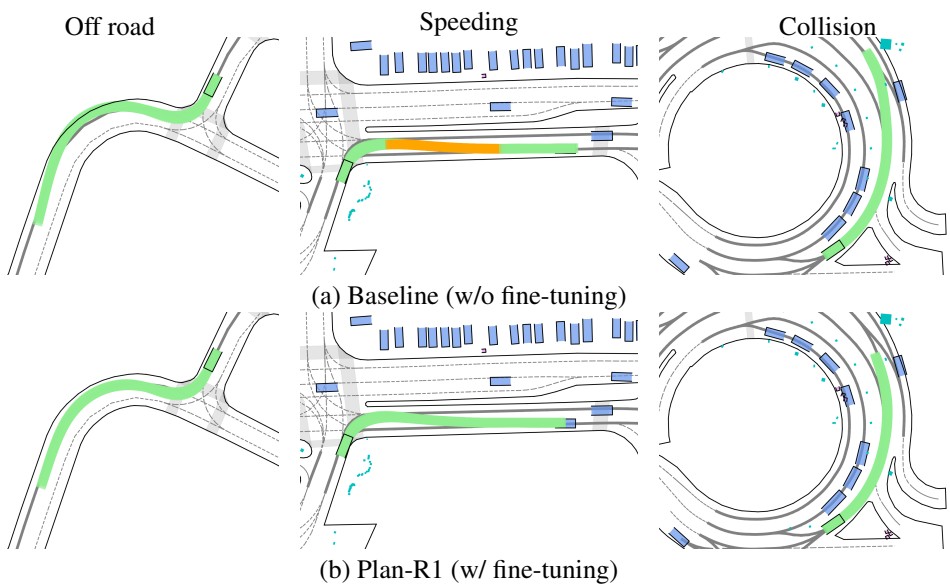

(a) Baseline (w/o fine-tuning)

(b) Plan-R1 (w/ fine-tuning)

Figure 3: Comparison of closed-loop ego trajectories generated by the pre-trained baseline (a) and Plan-R1 (b). Orange indicates speeding segments, and cyan marks static obstacles. The baseline exhibits issues such as off-road driving (left), speeding (middle), and collision with static obstacles (right), while Plan-R1 avoids these failures, producing safe and feasible trajectories.

preserves expert-like behavior. In the more challenging reactive setting, where surrounding agents dynamically respond to the ego vehicle, Plan-R1 achieves state-of-the-art scores of 87.69 (Val14), 77.20 (Test14-hard), and 90.04 (Test14-random), surpassing Diffusion Planner by +4.89, +7.98 and +7.11 points, respectively. This demonstrates Plan-R1's superior ability to handle highly interactive scenarios, thanks to the dual-model design and rule-based principle alignment. Following Diffusion Planner, we also apply the existing refinement module (Sun et al., 2024) for post-processing without any parameter tuning, where Plan-R1 still achieves the highest NR-CLS and R-CLS of 94.72 and 93.54 on Val14, respectively. These results validate the effectiveness of our two-stage framework: pre-training establishes a strong behavioral prior, while RL fine-tuning explicitly optimizes for safety and feasibility, enabling Plan-R1 to generate trajectories that are both robust and compliant.

**Qualitative Results.** We conduct a case study where the expert trajectory violates the local speed limit. As shown in Figure 2, both PLUTO (Cheng et al., 2024a) and Diffusion Planner (Zheng et al., 2025), trained solely on expert demonstrations, replicate this speeding behavior. In contrast, Plan-R1 maintains a compliant velocity throughout the rollout, demonstrating its ability to correct undesirable behaviors inherited from expert data. This highlights the effectiveness of rule-based RL fine-tuning in overcoming the limitations of expert-only training, producing more reliable behaviors.

## 4.3 ABLATION STUDIES

**The importance of rule-based RL fine-tuning.** We first evaluate the effect of adding rule-based RL fine-tuning to the pre-trained model. As shown in Table 2, GRPO fine-tuning consistently improves the overall score (e.g., +3.04 NR-CLS, +5.54 R-CLS) and most individual metrics (e.g.,

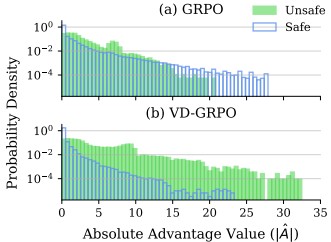

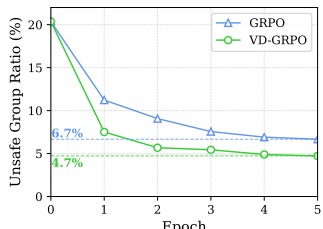

Table 3: Ablation on model capacity and world model (WM) choices.

| Capacity / WM | R-CLS |
|---|---|
| Pre-train | 82.81 |
| Pre-train ($2\times$) | 84.94 |
| GT replay | 87.44 |
| Reactive WM (Ours) | **90.04** |

Figure 4: Distributions of $|\hat{A}|$ for safe vs. unsafe groups.

Figure 5: Proportion of unsafe groups during training.

+2.29 drivable area compliance), showing that rule-based RL explicitly enhances safety awareness while also correcting undesirable behaviors, such as speeding, that are inherited from human demonstrations. Qualitative results in Figure 3 further illustrate this improvement: the pre-trained baseline exhibits off-road driving, speeding, and collision with static obstacle, while the RL fine-tuned model produces safe and feasible trajectories.

**The effect of VD-GRPO.** While standard GRPO significantly improves soft objectives such as progress (+2.47) and speed compliance (+3.08), it causes a drop of -0.96 in the critical collision avoidance metric, which is undesirable. To investigate this issue, we analyze the distributions of absolute advantage values ($|\hat{A}|$) for safe and unsafe (violation) groups, as shown in Figure 4 (a). Even though our reward function (Equation 3) assigns absolute priority to safety indicators such as collision avoidance via multiplicative terms, the two distributions almost completely overlap in the low-advantage region, while safe groups with small variance from soft-term fluctuations produce disproportionately large advantages, and unsafe groups with higher variance yield smaller advantages. This stems from GRPO's group-wise normalization, which erases reward scale differences across groups and dilutes rare yet critical safety signals.

As shown in Figure 4 (b), VD-GRPO addresses this issue by preserving absolute reward magnitudes, allowing safety-critical (unsafe) groups to naturally produce larger advantages and remain dominant throughout training. We further track the proportion of unsafe groups during training (Figure 5), where VD-GRPO reduces this ratio from 6.7% to 4.7% (29.8% reduction), indicating substantially safer planning behaviors. Results in Table 2 further confirm these benefits: collision avoidance +3.45, drivable area compliance +0.39, NR-CLS +2.58, and R-CLS +1.69 compared to GRPO. These findings demonstrate that VD-GRPO effectively amplifies rare yet critical safety signals, continuously improving safety while maintaining balanced performance on secondary objectives.

**Dual-model design.** We conduct an ablation study to verify the effectiveness of our dual-model design (Figure 3). Simply replaying logged trajectories of surrounding agents (GT replay) yields an R-CLS score of 87.44, which is better than the pre-trained model but substantially below our full method. This indicates that non-reactive simulation provides limited benefit, as it fails to capture realistic multi-agent interactions and thus cannot fully support effective reinforcement learning fine-tuning. Replacing GT replay with a learned world model substantially improves performance: our reactive world model achieves an R-CLS of 90.04, demonstrating the importance of interaction-aware simulation. To isolate this effect from model capacity, we also compare against a pre-trained baseline with doubled parameters (Pre-train ($2\times$)). While the larger model provides a modest gain of +2.13, it is far smaller than the +7.23 improvement achieved by introducing the reactive world model. This confirms that the performance gain primarily stems from our dual-model design rather than network size. These results highlight the critical role of modeling responsive surrounding-agent behaviors during RL fine-tuning, enabling stable optimization and superior closed-loop performance. Finally, more results and ablation studies can be found in Appendix A and Appendix B.

## 5 CONCLUSION

We presented Plan-R1, a two-stage framework that decouples planning principle alignment from behavior learning for safe and feasible trajectory planning. In the first stage, a motion predictor is

pre-trained on expert demonstrations to capture diverse, human-like driving behaviors. In the second stage, the ego policy is fine-tuned with rule-based rewards to explicitly align planning with principles such as safety, comfort, and traffic rule compliance. To address the limitation of standard GRPO, where group-wise normalization suppresses optimization for rare but critical safety violations, we proposed Variance-Decoupled GRPO (VD-GRPO), which preserves absolute reward magnitudes so that safety-critical objectives remain dominant throughout training. Experiments on the nuPlan benchmark demonstrate that Plan-R1 significantly enhances safety and feasibility, achieving state-of-the-art performance, particularly in challenging reactive settings.

## 6 ACKNOWLEDGEMENT

This work is supported by the National Natural Science Foundation of China (Nos.62495082, 62461160331, and 62495084).

## USE OF LARGE LANGUAGE MODELS (LLMS)

We acknowledge that large language models (LLMs) were used solely for minor manuscript editing and language polishing. No LLMs were involved in data collection, labeling, model training, or generating any experimental results. The scientific contributions, including algorithm design, implementation, and evaluation, were entirely completed by the authors.

## ETHICS STATEMENT

Our work focuses on improving the safety and reliability of autonomous vehicle planning. While improved planning algorithms have the potential to enhance road safety, premature deployment of this technology in real-world systems could lead to safety-critical failures. To mitigate these risks, we emphasize that our method is evaluated solely in simulation, and any real-world deployment should undergo rigorous validation and regulatory approval. We disclose all methodological details transparently to foster responsible research and enable auditing by the community.

## REPRODUCIBILITY STATEMENT

We have taken several steps to ensure the reproducibility of our work. Our Plan-R1 is described in detail in section 3. All hyperparameters, model architectures, and training configurations are fully reported in Appendix D. We release our source code, model checkpoints, and experiment scripts required to reproduce our results, which are available at `https://github.com/XiaolongTang23/Plan-R1.`. The nuPlan dataset used in our experiments is publicly available, and our data preprocessing pipeline is provided within the released codebase. In addition, we include comprehensive ablation studies in subsection 4.3 and Appendix B to verify the robustness of our approach.

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

# A  VISUALIZATION OF CLOSED-LOOP PLANNING RESULTS

Figure 6: Closed-loop planning results: each row represents a scenario at 0, 5, 10, and 15 seconds intervals. Each frame includes the future planning of the ego vehicle, predictions for neighboring agents.

# B  ADDITIONAL RESULTS

**Pass@$k$ analysis.**    To further assess the reliability of our method under multi-sampling conditions, we conduct a pass@$k$ (Kulal et al., 2019) analysis which measures the probability that at least one out of $k$ generated samples is successful. In our context, a "successful" trajectory is defined as one that satisfies all core safety and feasibility criteria, including staying within drivable areas, avoiding collisions, respecting speed limits, and maintaining comfort. For both the pretrained baseline and our Plan-R1, we generate $k$ trajectories per scenario and compute the proportion of cases where at least one trajectory meets these criteria. As shown in Figure 7, pass@$k$ improves with increasing $k$ for both models. However, Plan-R1 consistently outperforms the baseline, indicating it yields high-quality trajectories with greater likelihood. This suggests that fine-tuning not only improves

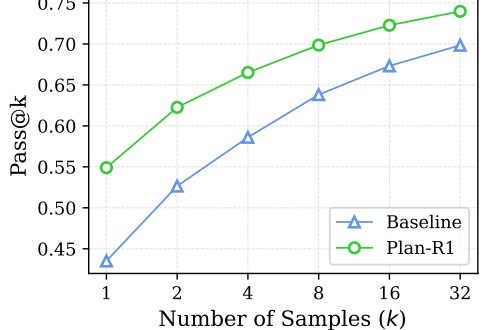

Table 4: Ablation on Group Size $G$.

| $G$ | NR-CLS | R-CLS | GPU Memory (GB) |
|---|---|---|---|
| 2 | 90.60 | 89.81 | 12 |
| 4 (used) | **91.23** | **90.04** | 24 |
| 6 | 91.12 | 89.35 | 36 |

Figure 7: Comparison of pass@$k$ between the baseline and Plan-R1.

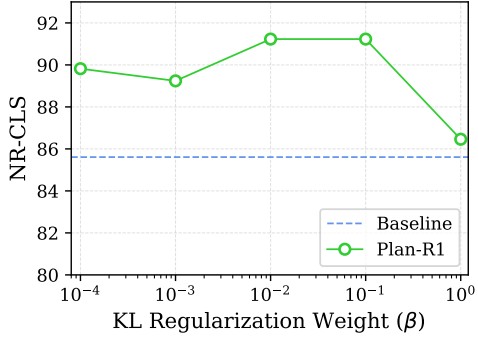

Figure 8: Effect of KL regularization weight.

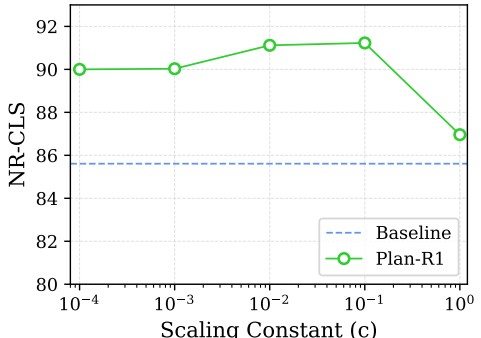

Figure 9: Effect of fixed scaling constant.

the single-shot performance but also enhances the model's ability to generate feasible trajectories across multiple samples, which is essential for robust decision-making in safety-critical domains where uncertainty must be mitigated through diverse sampling.

**Ablation study on group size.** We evaluate the impact of different group sizes ($G$), where $G$ determines the number of sampled trajectories per scenario for computing group-level advantages. As shown in Figure 4, increasing $G$ from 2 to 4 improves both NR-CLS and R-CLS. However, further increasing $G$ to 6 yields no performance gain and causes a significant rise in GPU memory usage (from 24GB to 36GB). Thus, we set $G = 4$ in all main experiments, achieving the best balance between accuracy and computational efficiency.

**Effect of KL regularization and fixed scaling.** We study two key hyperparameters in VD-GRPO: KL Regularization Weight ($\beta$) and fixed Scaling Constant ($c$). $\beta$ controls how closely the fine-tuned policy stays aligned with the pretrained reference policy. As shown in Figure 8, setting $\beta$ too low causes the model to deviate excessively, harming generalization and stability. Conversely, an excessively high $\beta$ prevents the policy from improving safety-critical behaviors. A moderate value ($\beta = 0.1$) achieves the best trade-off between exploration and behavioral retention. The scaling constant $c$ in VD-GRPO (Equation 5) controls the magnitude of the unnormalized rewards. As shown in Figure 9, performance is stable within a wide range ($10^{-4} \leq c \leq 10^{-1}$). Setting $c$ too low amplifies gradient noise, while too high a value diminishes the effect of fine-tuninig. In all experiments, we fix $c = 10^{-1}$ for stable and efficient optimization.

Notably, these two hyperparameters interact in a competitive manner: $\beta$ governs the strength of the KL regularization loss, which encourages the policy to stay close to the pretrained reference, while $c$ determines the relative strength of the reinforcement learning signal that drives policy improve-

Table 5: Ablation study on the reward components.

| Reward setting | NR-CLS | Collision | TTC | Drivable | Speed | Comfort | Progress |
|---|---|---|---|---|---|---|---|
| w/o Collision | 73.10 | 94.15 | 74.33 | 96.56 | 99.55 | 99.23 | 94.15 |
| w/o Drivable | 88.88 | 96.17 | 94.25 | 95.79 | 99.40 | 99.62 | 92.19 |
| w/o Speed | 90.04 | 94.83 | 92.29 | 98.47 | 92.29 | 99.62 | 98.02 |
| w/o Comfort | 90.83 | 96.93 | 93.87 | 97.32 | 99.47 | 99.23 | 93.16 |
| w/o Progress | 80.01 | 97.70 | 95.79 | 98.85 | 99.69 | 98.85 | 68.95 |
| Full rewards (Plan-R1) | 91.23 | 97.32 | 95.02 | 97.32 | 99.45 | 99.62 | 91.94 |

ment. A higher $\beta$ implicitly suppresses the influence of the RL objective, making the optimization more conservative, whereas a larger $c$ amplifies the RL gradients and pushes the policy toward more aggressive exploration and deviation from the reference. Thus, tuning these two hyperparameters involves a delicate trade-off between stability and exploration, where $\beta$ ensures retention of safe, human-like behaviors, while $c$ enables sufficient optimization pressure to achieve safety-critical improvements.

**Ablation study on reward components.**   We further investigate the contribution of different reward components by removing each term from the total reward function in Equation 3. As shown in Table 5, each component plays a distinct role:

- Collision term is the most critical: removing it causes NR-CLS to drop sharply to 73.10, demonstrating that explicit collision avoidance is essential for safe planning.

- Drivable area compliance has a strong effect on feasibility, with removal leading to noticeable decrease in Drivable metric.

- Speed limit compliance and Comfort terms provide secondary benefits, encouraging smoother and rule-compliant driving behaviors.

- Progress term strongly affects trajectory efficiency: without it, the model tends to stay stationary or overly conservative, resulting in a low NR-CLS of 80.01.

These results confirm that our reward design balances safety and efficiency, with each component playing a complementary role.

**Results on interPlan.**   To further assess the robustness of our Plan-R1 under challenging and perturbed scenarios, we additionally evaluate the model on the interPlan benchmark Hallgarten et al. (2024) under all 335 scenarios. InterPlan has recently been adopted by some planning works Sun et al. (2024) as an auxiliary robustness testbed. It is derived from nuPlan but systematically perturbs the original scenes by adding or removing agents, inserting obstacles, or modifying goals to construct long-tail, interaction-heavy situations that stress-test a planner's generalization capability.

As shown in Table 6, without post-processing, Plan-R1 achieves a score of 56.64, slightly below PLUTO (57.74) but outperforming all other baselines. This is expected because interPlan perturbs scenes by adding or removing agents and obstacles—an augmentation strategy that aligns closely with PLUTO's contrastive training pipeline, naturally favoring it. With post-processing, Plan-R1* achieves the highest overall score (72.33), outperforming PDM-Closed* by +2.69. These results demonstrate that Plan-R1 exhibits strong robustness under perturbed and long-tail scenarios, further validating its effectiveness beyond the standard nuPlan benchmark.

**Robustness of the Frozen World Model Under Ego-State Perturbations.**   A key requirement of our dual-model rollout is that the frozen world model must remain reliable even when the ego policy deviates from expert demonstrations. If the world model were highly sensitive to ego-distribution shift, inaccuracies in surrounding-agent predictions could accumulate during rollouts and introduce compounding errors, weakening the effectiveness of RL fine-tuning. To assess this potential issue, we conduct an additional robustness experiment that evaluates the stability of the frozen world model under controlled perturbations of the ego state. Specifically, we inject zero-mean Gaussian noise

Table 6: Comparison with SOTAs on interPlan benchmark. The best result is in **bold** and the second best result is underlined. *: with rule-based post-processing.

| Type | Planner | interPlan |
|---|---|---|
| Expert | Log-Replay | 14.76 |
| Rule-based & Hybrid | IDM (Treiber et al., 2000) | 47.07 |
| | PDM-Closed* (Dauner et al., 2023) | 69.64 |
| | PLUTO* (Cheng et al., 2024a) | 63.88 |
| | Plan-R1* (Ours) | **72.33** |
| Learning-based | UrbanDriver (Scheel et al., 2022) | 5.56 |
| | PDM-Open (Dauner et al., 2023) | 26.22 |
| | PlanTF (Cheng et al., 2024b) | 47.72 |
| | PLUTO (Cheng et al., 2024a) | **57.74** |
| | Diffusion Planner (Zheng et al., 2025) | 50.07 |
| | Plan-R1 (Ours) | 56.64 |

of increasing magnitude ($\sigma = 0, 0.1, 0.3, 1, 3, 10m$) into the ego's observed historical and current states, and measure the resulting prediction errors for surrounding agents using ADE and FDE.

As shown in Table 7, the frozen world model is highly stable under ego-distribution shifts. For realistic ego deviations ($\sigma < 1m$), degradation is minimal: ADE increases by only +0.11 m and FDE by +0.34 m. For larger perturbations ($\sigma \geq 1m$), errors quickly saturate rather than diverge. Even under an extreme perturbation of $\sigma = 10m$ (far beyond any plausible deviation), the FDE increases by only +0.58 m, with no indication of exploding or compounding drift. This confirms that the world model provides reliable surrounding-agent trajectories even when the ego input is significantly distorted. This robustness arises because the world model conditions on multiple complementary cues—each agent's own motion history, interaction context, and map topology—of which the ego state is only one component. As a result, moderate ego deviations do not dominate the multi-agent representation, and even extreme deviations are effectively anchored by the consistent behavior of surrounding agents and road structure, preventing instability during multi-step rollouts.

Table 7: Robustness of the frozen world model under ego-state perturbations. Gaussian noise with standard deviation $\sigma$ is injected into ego states.

| Noise $\sigma$ (m) | ADE (m) $\downarrow$ | FDE (m) $\downarrow$ |
|---|---|---|
| 0 | 1.03 | 3.01 |
| 0.1 | 1.07 | 3.15 |
| 0.3 | 1.14 | 3.35 |
| 1.0 | 1.22 | 3.53 |
| 3.0 | 1.25 | 3.59 |
| 10.0 | 1.26 | 3.59 |

## C  RULE-BASED REWARD DESIGN

We design the total reward as a product of two components: a set of binary safety indicators and a weighted sum of soft cost terms. This formulation first ensures that no reward is given if any critical safety condition is violated, and then encourages the agent to optimize for smoother and more efficient trajectories when safety is satisfied. Below, we describe the components in detail.

**Safety indicators ($\mathcal{I}_{\text{safe}}$).**  The following constraints must all be satisfied to receive a non-zero reward:

- **Driving area compliance**: All predicted positions must lie within the drivable area polygon.
- **No collision with dynamic agents**: At each prediction step, the ego vehicle must not collide with the predicted positions of any surrounding agent at the corresponding time step.

- **No collision with static obstacles**: All predicted positions must avoid collisions with static obstacles.

To ensure accurate safety assessment, all collision checks are performed using the full bounding boxes of the ego vehicle and surrounding agents, rather than using center points.

**Soft cost terms ($\mathcal{I}_{\mathbf{cost}}$).** When all safety indicators are satisfied, the agent receives a soft reward computed as a weighted sum of four normalized terms:

- **Comfort** ($w = 2$): This term penalizes high lateral and longitudinal accelerations, angular velocity, and angular acceleration. If any of these exceed predefined thresholds, the comfort score is set to 0; otherwise, it is 1.
- **Time-to-collision (TTC)** ($w = 5$): This term is set to 1 if the predicted TTC remains above a safety threshold; otherwise, it is set to 0.
- **Speed limit compliance** ($w = 2$): This term linearly penalizes the predicted speeding behavior by computing the ratio between the total over-speed distance and a predefined maximum over-speed threshold. The score decreases linearly from 1 to 0 as this ratio increases, following the same formulation as in the nuPlan benchmark (Caesar et al., 2021).
- **Progress** ($w = 1$): This term measures the forward progress of the predicted trajectory along the reference route. It is normalized using the progress of the expert trajectory. The progress is computed by projecting each predicted position onto the centerline of the reference route and accumulating the forward distance along the projected path. Note that this is not an imitation loss: we do not minimize the distance between the predicted and expert trajectories, but only use the expert's final progress as a normalization reference.

Importantly, most reward components, such as collisions, comfort, and TTC, are naturally defined at the token level since they depend on each predicted position or step. However, *Speed limit compliance* and *Progress* are inherently trajectory-level metrics because they require aggregated information over the entire predicted trajectory (e.g., total distance traveled or cumulative overspeed distance). To integrate these trajectory-level metrics into token-level GRPO optimization, we simply normalize each trajectory-level reward to $[0, 1]$ and assign the same value to every token within the trajectory. This ensures that all tokens share identical signals for these terms, while maintaining consistent token-level advantage computation in GRPO. Empirically, this straightforward approach proved stable and effective, providing clear optimization signals without introducing additional variance.

## D  IMPLEMENTATION DETAILS

**Architecture.** We use a discrete motion token vocabulary of size 1024 for each agent category (Vehicle, Pedestrian, Cyclist), where each token corresponds to a 0.5-second movement segment. The ego agent shares the same token vocabulary as other vehicles in the Vehicle category. Our model adopts a 6-layer Transformer decoder, each with 8 attention heads and a hidden dimension of 128. At the end of each decoder layer, a token prediction head is applied, implemented as a two-layer MLP. The parameters of an autoregressive predictor are approximately 5.05M, and total parameters of dual-model are about 10.1M.

**Training.** During pretraining, we use teacher forcing to optimize next-motion-token prediction with cross-entropy loss. The model is trained for 32 epochs with a batch size of 64 using the AdamW optimizer (Loshchilov & Hutter, 2018), with a learning rate of $3 \times 10^{-4}$ and a weight decay of $1 \times 10^{-4}$. A dropout rate of 0.1 is applied throughout training. A cosine annealing scheduler is used to decay the learning rate to zero during pretraining. For fine-tuning, we apply GRPO with a group size of $G = 4$, KL divergence regularization weight $\beta = 0.1$, and a reduced learning rate of $4 \times 10^{-6}$. Fine-tuning is performed for 5 epochs, using the same dropout rate of 0.1 as in pretraining. No data augmentation is applied during either training or fine-tuning. All experiments are conducted on 8 NVIDIA RTX 4090 GPUs.

**Inference.** During closed-loop simulation, both the pretrained and fine-tuned models generate trajectories by selecting the top-1 token at each step.

# E  THEORETICAL ANALYSIS OF VD-GRPO

In this section, we provide a complete theoretical analysis of VD-GRPO. We focus on two questions: (i) why GRPO's group-wise variance normalization is problematic in safety-critical multi-objective planning, and (ii) why introducing a fixed scaling constant $c$ after removing variance is theoretically sound and practically necessary. We follow the notations in Section 3.

## E.1  PRELIMINARIES

For each scenario with context $C$ and planning principles $P$, we sample a group of $G$ future trajectories $\{Y^1, \ldots, Y^G\}$ from the old ego policy $\pi_{e_{\text{old}}}$. Each trajectory $Y^g = \{y_1^g, \ldots, y_F^g\}$ has horizon $F$ and receives token-level rewards $R(y_t^g)$. The fine-tuning objective (Equation (4)) is:

$$\mathcal{L}_{\text{finetune}} = -\frac{1}{GF} \sum_{g=1}^{G} \sum_{t=1}^{F} \left[ \frac{\pi_e(y_t^g|C, P, y_{<t}^g)}{\pi_{e_{\text{old}}}(y_t^g|C, P, y_{<t}^g)} \hat{A}_t^g - \beta \, D_{\text{KL}}[\pi_e \,\|\, \pi_{\text{ref}}] \right], \tag{6}$$

where $\hat{A}_t^g$ is the cumulative advantage of rollout $g$ from step $t$:

$$\hat{A}_t^g = \sum_{\tau=t}^{F} \tilde{R}(y_\tau^g). \tag{7}$$

Throughout the analysis, we treat the KL term as an independent regularizer: its gradient is additive and does not interact with reward normalization. Thus, to analyze the effect of normalization, we focus on the RL term.

## E.2  GRPO GRADIENT AND THE IMPLICIT VARIANCE WEIGHT

**Group-wise normalization in GRPO.**  Standard GRPO normalizes rewards within each sampled group:

$$\tilde{R}^{\text{GRPO}}(y_t^g) = \frac{R(y_t^g) - \mu_R}{\sigma_R}, \tag{8}$$

where

$$\mu_R = \frac{1}{F} \sum_{t=1}^{F} R(y_t^g), \qquad \sigma_R = \sqrt{\frac{1}{F} \sum_{t=1}^{F} (R(y_t^g) - \mu_R)^2}. \tag{9}$$

Plugging Equation (8) into Equation (7) and then into Equation (6), the RL-only gradient of GRPO becomes:

$$\nabla_\theta \mathcal{L}_{\text{GRPO}} = -\frac{1}{GF} \sum_{g,t} \left( \sum_{\tau=t}^{F} \tilde{R}^{\text{GRPO}}(y_\tau^g) \right) \nabla_\theta \log \pi_e \tag{10}$$

$$= -\frac{1}{GF} \sum_{g,t} \left( \frac{1}{\sigma_R^{(g)}} \sum_{\tau=t}^{F} (R(y_\tau^g) - \mu_R^{(g)}) \right) \nabla_\theta \log \pi_e. \tag{11}$$

**Key observation.**  Equation (11) shows that for each group $g$, the entire advantage is scaled by $1/\sigma_R$. Hence GRPO implicitly applies a *group-level importance weight*

$$w_{\text{GRPO}} = \frac{1}{\sigma_R}. \tag{12}$$

In single-objective tasks (e.g., mathematical reasoning), this weighting helps equalize gradient magnitudes across groups. However, in multi-objective planning where reward scales encode priorities, such variance-based reweighting can distort optimization.

### E.3 REWARD STRUCTURE INDUCES LARGER VARIANCE FOR UNSAFE GROUPS

Our total reward (Equation (3)) is:

$$R(y_t^g) = \prod_{k \in \mathcal{I}_{\text{safe}}} \mathbf{1}_{k,t,g} \cdot \sum_{j \in \mathcal{I}_{\text{cost}}} w_j \, r_j(y_t^g), \tag{13}$$

where $\mathbf{1}_{k,t,g} \in \{0,1\}$ is a multiplicative safety indicator (e.g., collision avoidance, drivable area compliance), and $\sum_j w_j r_j(y_t^g)$ is a weighted sum of soft cost terms (e.g., comfort, speed compliance, progress).

Define

$$S_t^g \triangleq \prod_{k \in \mathcal{I}_{\text{safe}}} \mathbf{1}_{k,t,g} \in \{0,1\}, \qquad C_t^g \triangleq \sum_{j \in \mathcal{I}_{\text{cost}}} w_j \, r_j(y_t^g) \in [0,1], \tag{14}$$

so that $R(y_t^g) = S_t^g \, C_t^g$.

**Safe vs. unsafe groups.** A safe group is one where $S_t^g = 1$ for all $t$ and all rollouts $g$, so rewards are purely soft-cost:

$$R(y_t^g) = C_t^g \quad \text{(safe group)}. \tag{15}$$

An unsafe group contains at least one step where $S_t^g = 0$, so some rewards are nullified to exactly zero:

$$R(y_t^g) = 0 \text{ for some } t, g, \quad R(y_t^g) = C_t^g \text{ for others (unsafe group)}. \tag{16}$$

**Lemma E.1** (Variance bound for distributions on the unit interval). *For any random variable $X \in [0,1]$, its variance satisfies*

$$\text{Var}(X) \leq \tfrac{1}{4}. \tag{17}$$

*Proof.* Let $\mu = \mathbb{E}[X] \in [0,1]$. Since $0 \leq X \leq 1$, we have $X^2 \leq X$, hence $\mathbb{E}[X^2] \leq \mathbb{E}[X] = \mu$. Therefore, $\text{Var}(X) = \mathbb{E}[X^2] - \mu^2 \leq \mu - \mu^2 = \mu(1-\mu)$. The quadratic function $\mu(1-\mu)$ is maximized at $\mu = \frac{1}{2}$, giving $\text{Var}(X) \leq \frac{1}{4}$. Equality holds iff $X \in \{0,1\}$ a.s. and $\Pr(X = 1) = \Pr(X = 0) = \frac{1}{2}$. $\square$

**Lemma E.2** (Variance of unsafe reward mixture). *Consider a non-trivial unsafe group where a fraction $\alpha \in (0,1)$ of rewards are zero and the remaining fraction $(1 - \alpha)$ are positive soft-cost rewards $C_t^g$. Then the unsafe-group reward satisfies*

$$\text{Var}(R_t^g) = (1-\alpha)\text{Var}(C_t^g) + \alpha(1-\alpha)\big(\mathbb{E}[C_t^g]\big)^2. \tag{18}$$

*Proof.* Let

$$R_t^g = \begin{cases} 0, & \text{w.p. } \alpha, \\ C_t^g, & \text{w.p. } 1-\alpha. \end{cases}$$

Then $\mathbb{E}[R_t^g] = (1-\alpha)\mathbb{E}[C_t^g]$ and $\mathbb{E}[(R_t^g)^2] = (1-\alpha)\mathbb{E}[(C_t^g)^2]$. Using $\text{Var}(X) = \mathbb{E}[X^2] - \mathbb{E}[X]^2$ and $\mathbb{E}[C^2] = \text{Var}(C) + (\mathbb{E}[C])^2$ yields Eq. (E.1). $\square$

**Corollary E.1** (Unsafe groups have larger variance under task statistics). *Assume: (i) $C_t^g \in [0,1]$ with $\text{Var}(C_t^g) \leq 1/4$ (Lemma E.1); (ii) unsafe groups satisfy $\alpha < 0.3$; and (iii) $\mathbb{E}[C_t^g] > 0.8$ (empirically measured under the pretrained model). Then unsafe groups have strictly larger reward variance:*

$$\text{Var}(R_t^g) > \text{Var}(C_t^g). \tag{19}$$

*Proof.* From Lemma E.2,

$$\text{Var}(R_t^g) - \text{Var}(C_t^g) = \alpha\Big[(1-\alpha)(\mathbb{E}[C_t^g])^2 - \text{Var}(C_t^g)\Big].$$

Using $\alpha < 0.3$ and $\mathbb{E}[C_t^g] > 0.8$ gives $(1-\alpha)(\mathbb{E}[C_t^g])^2 > 0.7 \times 0.64 = 0.448$. Meanwhile Lemma E.1 gives $\text{Var}(C_t^g) \leq 0.25$. Thus the bracketed term in (E.2) exceeds 0.198, implying $\text{Var}(R_t^g) - \text{Var}(C_t^g) > 0$. $\square$

**Consequence for GRPO.** By Corollary E.1 and the GRPO weight in Equation (12), unsafe groups satisfy $\sigma_R(\text{unsafe}) > \sigma_R(\text{safe})$, hence

$$w_{\text{GRPO}}(\text{unsafe}) < w_{\text{GRPO}}(\text{safe}). \tag{20}$$

Therefore GRPO systematically down-weights safety-critical trajectories, contrary to the intended priority structure in planning.

### E.4 VD-GRPO REMOVES VARIANCE-INDUCED BIAS

VD-GRPO replaces Equation (8) with centering and fixed scaling:

$$\tilde{R}^{\text{VD}}(y_t^g) = \frac{R(y_t^g) - \mu_R}{c}, \tag{21}$$

where $c > 0$ is a global constant shared by all groups.

Plugging Equation (21) into Equation (6), the RL-only gradient becomes

$$\nabla_\theta \mathcal{L}_{\text{VD}-\text{GRPO}} = -\frac{1}{cGF} \sum_{g,t} \left( \sum_{\tau=t}^{F} (R(y_\tau^g) - \mu_R) \right) \nabla_\theta \log \pi_e. \tag{22}$$

**Key property.** Equation (22) shows that VD-GRPO applies the same scaling to all groups, thus removing the implicit variance weight $1/\sigma_R$. Gradient magnitudes therefore reflect true reward differences: safe groups (only soft costs, small fluctuations) contribute smaller gradients, while unsafe groups retain larger gradients consistent with higher safety priority.

### E.5 ROLE OF THE SCALING CONSTANT $c$

#### E.5.1 THEORETICAL SOUNDNESS: $c$ DOES NOT CHANGE OPTIMALITY

From Equation (22), introducing $c$ multiplies all advantages by the same positive scalar $1/c$. Hence the update direction, stationary points, and optimal policies are unchanged.

#### E.5.2 PRACTICAL NECESSITY: RESTORING RL–KL BALANCE

Because $R(y_t^g) \in [0, 1]$, the within-group deviation is bounded.

**Lemma E.3** (Upper bound on $\sigma_R$). *For any group whose rewards lie in $[0, 1]$, we have*

$$0 < \sigma_R \leq \frac{1}{2}.$$

*Proof.* By Lemma E.1, $\text{Var}(R_t^g) \leq 1/4$ for any bounded reward, so $\sigma_R \leq 1/2$. $\square$

Thus GRPO implicitly amplifies RL advantages by

$$\frac{1}{\sigma_R} \geq 2. \tag{23}$$

The KL coefficient $\beta$ in Equation (6) was tuned under this implicit amplification. If we remove $\sigma_R$ without compensation, the RL gradient shrinks relative to $\beta \text{KL}(\cdot)$, causing KL to dominate and suppress policy improvement. VD-GRPO introduces $c \leq 1$ to restore the RL scale to the regime implicitly expected by GRPO, while still avoiding variance-induced bias. This explains the stable performance across a wide range of $c$ observed in Figure 9.

### E.6 SUMMARY

GRPO's variance normalization induces an implicit group weight $1/\sigma_R$. Under our reward design, unsafe groups exhibit larger reward variance than safe groups, so GRPO down-weights safety-critical trajectories. VD-GRPO removes this bias by decoupling normalization from variance. The fixed scaling constant $c$ is theoretically benign (does not alter optimality) and practically required to preserve the RL–KL balance without retuning $\beta$.

