# OpenReview forum: "Plan-R1: Safe and Feasible Trajectory Planning as Language Modeling"
_ICLR.cc/2026/Conference — ICLR 2026 Poster_

### Official Review · Reviewer_fE6E · 2025-10-17

**Soundness:** 3
**Presentation:** 3
**Contribution:** 3
**Rating:** 6
**Confidence:** 3

**Summary:**

This paper studies how to use RL with rule-based rewards like GRPO to align ego planning with principles for trajectory planning models. Overall, the authors utilize a two-stage paradigm: pretraining and post-training to align with rules. They also modify GRPO into VD-GRPO, so as to remain safety-critical objectives remain dominant. Experiments are conducted on nuPlan dataset, and advantage against other previous methods are shown.

**Strengths:**

1. The authors study the interesting and important problem on whether recent advances in post-training language model, or basically GRPO, can help improve trajectory planning models. Their observations and modifications on GRPO provide clear and inspiring answer to this question. Especially, they conduct ablation study, analyzing the dominating term of advantages, and show that during GRPO training the unsafe samples are not assigned enough importance.
2. Experiments on nuPlan prove the validity of their proposed method.
3. The dual-model rollout, which utilizes frozen pre-trained model as 'world model' for other driving agents, enables the trained model to interact with other driving agents. This is also an interesting design choice.
Overall, the discovery about GRPO is interesting, and related improvement method, i.e. VD-GRPO, is reasonable and shown to be effective.

**Weaknesses:**

1. Experiments are mainly conducted on nuPlan, not including other datasets. Given that said, experiments are thorough on nuPlan.

**Questions:**

Have you tried to use your fine-tuned model as 'world model' for other driving agents? Though I indeed recognize that this might not give too much performance uplift.

---

> ### Author Response · Authors · 2025-11-22
> **Response to Reviewer fE6E 1/2**
>
> We thank the reviewer for the constructive comments and positive feedback on our paper. Regarding the concerns of the reviewer fE6E, we provide the following responses.
>
> > **W1**: Experiments are mainly conducted on nuPlan, not including other datasets. Given that said, experiments are thorough on nuPlan.
>
> We appreciate the reviewer's constructive comments. **Our experiments focus on nuPlan because it is currently the only public benchmark that offers both closed-loop multi-agent simulation and planning-level evaluation metrics, which are essential for evaluating a trajectory planner.** Other datasets unfortunately do not meet these requirements:
> - Argoverse / Argoverse 2: only open-loop prediction metrics with no closed-loop evaluators; prior work [1] shows that open-loop accuracy correlates weakly with planning quality.
> - Waymo Open Motion Dataset: supports motion prediction and sim agents, but lacks planning-level evaluators.
> - CARLA / nuScenes / NAVSIM: primarily used for end-to-end driving, requiring perception and control stacks and not directly compatible with planning models.
>
> Consequently, nearly all recent planning works, such as PlanTF [2], CarPlanner [3], GenDrive [4], and PLUTO [5], report results only on nuPlan. Diffusion Planner [6] additionally reports results on a private dataset. As the reviewer noted, **we already conduct extensive experiments on three representative nuPlan splits (Val14, Test14-random, Test14-hard) and under both reactive and non-reactive closed-loop settings, consistently achieving state-of-the-art performance.**
>
> To further address the reviewer’s concern, we additionally evaluate Plan-R1 on the interPlan [7] benchmark, which has recently been adopted by several planning works [8,9] as an auxiliary testbed for robustness. InterPlan is derived from nuPlan but systematically perturbs scenes (adding/removing agents, injecting obstacles, altering goals) to create challenging long-tail and interaction-heavy scenarios. The results are as follows (* indicates with rule-based post-processing; **bold** denotes best, *italic* denotes second best):
>
> |Type|Planner|Score|
> |-|-|-|
> |Expert|log-play|14.76|
> |Learning-based|UrbanDriver|5.56|
> ||PDM-Open|26.22|
> ||PlanTF|47.72|
> ||Diffusion Planner|50.07|
> ||PLUTO| **57.74**|
> ||Plan-R1 (Ours)|*56.64*|
> |Rule-based & Hybrid| IDM*|47.07|
> ||PLUTO*|63.88 |
> ||PDM-Closed*|*69.64*|
> ||Plan-R1* (Ours)|**72.33**|
>
> - Without post-processing, Plan-R1 achieves 56.64, slightly below PLUTO (57.74) but outperforming all other baselines by a clear margin. This is expected because interPlan constructs scenes by adding or removing agents and obstacles—an augmentation strategy closely aligned with PLUTO’s contrastive learning pipeline, naturally favoring it.
> - With post-processing, Plan-R1* achieves the best overall performance (72.33), outperforming PDM-Closed* by +2.69, demonstrating strong robustness in perturbed and long-tail scenarios.

---

> > ### Author Response · Authors · 2025-11-22
> > **Response to Reviewer fE6E 2/2**
> >
> > > **Q1**: Have you tried to use your fine-tuned model as 'world model' for other driving agents? Though I indeed recognize that this might not give too much performance uplift.
> >
> > Thank you for the constructive question. We evaluated whether the RL-fine-tuned model can serve as the world model under the test14-random non-reactive closed-loop setting:
> > - Fine-tuned model as world model + pretrained model as planner initialization: 89.02
> > - Fine-tuned model as world model + fine-tuned model as planner initialization: 90.40
> > - Pretrained model as world model + pretrained model as planner initialization (default): 91.23
> >
> > Both substitutions perform worse than the default. The reason is that the pretrained model is a joint predictor for ego and surrounding agents, whereas RL fine-tuning updates parameters only through ego-centric rewards, degrading its ability to predict other agents. This degradation is directly measurable:
> > - Before fine-tuning: ADE 1.03, FDE 3.01
> > - After fine-tuning: ADE 1.26, FDE 3.67
> >
> > Using this degraded predictor as the world model results in less accurate and less reactive surrounding-agent behavior, reducing interaction realism and introducing compounding rollout errors. Thus, although feasible, using the fine-tuned model as the world model is not desirable; the frozen pretrained model provides more stable and accurate multi-agent predictions.
> >
> > **References**
> > [1] Daniel Dauner, Marcel Hallgarten, Andreas Geiger, and Kashyap Chitta. Parting with misconceptions about learning-based vehicle motion planning. In Conference on Robot Learning (CoRL),2023.
> > [2] Jie Cheng, Yingbing Chen, Xiaodong Mei, Bowen Yang, Bo Li, and Ming Liu. Rethinking imitationbased planners for autonomous driving. In IEEE International Conference on Robotics and Automation (ICRA), 2024.
> > [3] Dongkun Zhang, Jiaming Liang, Ke Guo, Sha Lu, Qi Wang, Rong Xiong, Zhenwei Miao, and Yue Wang. Carplanner: Consistent auto-regressive trajectory planning for large-scale reinforcement learning in autonomous driving. arXiv preprint arXiv:2502.19908, 2025.
> > [4] Huang, Zhiyu, Xinshuo Weng, Maximilian Igl, Yuxiao Chen, Yulong Cao, Boris Ivanovic, Marco Pavone, and Chen Lv. Gen-drive: Enhancing diffusion generative driving policies with reward modeling and reinforcement learning fine-tuning. In IEEE International Conference on Robotics and Automation (ICRA), 2025.
> > [5] Jie Cheng, Yingbing Chen, and Qifeng Chen. Pluto: Pushing the limit of imitation learning-based planning for autonomous driving. arXiv preprint arXiv:2404.14327, 2024.
> > [6] Yinan Zheng, Ruiming Liang, Kexin Zheng, Jinliang Zheng, Liyuan Mao, Jianxiong Li, Weihao Gu, Rui Ai, Shengbo Eben Li, Xianyuan Zhan, et al. Diffusion-based planning for autonomous driving with flexible guidance. arXiv preprint arXiv:2501.15564, 2025.
> > [7] Hallgarten, Marcel, Julian Zapata, Martin Stoll, Katrin Renz, and Andreas Zell. Can vehicle motion planning generalize to realistic long-tail scenarios?. In IEEE/RSJ International Conference on Intelligent Robots and Systems (IROS), 2024.
> > [8] Sun, Qiao, Huimin Wang, Jiahao Zhan, Fan Nie, Xin Wen, Leimeng Xu, Kun Zhan, Peng Jia, Xianpeng Lang, and Hang Zhao. Generalizing motion planners with mixture of experts for autonomous driving. In IEEE International Conference on Robotics and Automation (ICRA), 2025.
> > [9] Wang, Lingguang, Ömer Şahin Taş, Marlon Steiner, and Christoph Stiller. FlowDrive: moderated flow matching with data balancing for trajectory planning. arXiv preprint arXiv:2509.21961, 2025.

---

> > > ### Comment · Reviewer_fE6E · 2025-11-23
> > >
> > > I would like to thank the authors for their comprehensive and detailed rebuttal.
> > >
> > > More experiments and analysis on interPlan have shown the effectiveness of their proposed methods, and the ablation study with respect to world model and planner clearly shows the intuition and effectiveness of the authors' designs.
> > >
> > > Based on these, my concerns have been resolved, and I would like to increase my score from 6 to 8.

---

> > > > ### Author Response · Authors · 2025-11-24
> > > >
> > > > Thank you for the thoughtful follow-up and the positive evaluation.  We appreciate your constructive feedback, and we are glad that the additional experiments and analysis addressed your concerns. We will incorporate the improvements into the final version. Thanks again for your careful review!

---

### Official Review · Reviewer_5bV5 · 2025-10-26

**Soundness:** 3
**Presentation:** 4
**Contribution:** 3
**Rating:** 8
**Confidence:** 4

**Summary:**

This paper proposes Plan-R1, a two-stage trajectory planning framework for autonomous driving. A motion predictor is first pre-trained to capture diverse human-like driving behaviors, and then fine-tuned with rule-based reinforcement learning to align with safety and planning principles. The authors further identify a critical limitation in GRPO's normalization for safety-critical domains and introduce Variance-Decoupled GRPO (VD-GRPO) to preserve absolute reward magnitudes. Combined with a dual-model interactive rollout design, Plan-R1 achieves state-of-the-art performance on the nuPlan benchmark, especially under reactive closed-loop evaluation.

**Strengths:**

1. The authors clearly diagnose how per-group variance normalization in GRPO suppresses rare but safety-critical violations, and propose a principled solution that consistently enhances safety optimization without sacrificing secondary objectives.

2. Using a frozen world model for surrounding-agent responses ensures interaction-aware rollouts while preventing instability in non-ego behaviors. This design proves crucial for strong performance in reactive scenarios.

3. Plan-R1 achieves new state-of-the-art performance in both non-reactive and reactive nuPlan evaluations, with significant gains in safety metrics such as collisions and drivable area compliance. The ablations convincingly support the contributions of both VD-GRPO and the dual-model rollout.

**Weaknesses:**

1. Limited analysis of world model reliability when ego deviates from expert behavior. The frozen world model is assumed to remain accurate when the ego policy explores beyond regions well-covered by expert data. However, there is no case study or quantitative analysis showing how the world model behaves under large ego deviations or unusual interaction patterns, which could introduce compounding errors during RL fine-tuning.

2. Tokenization design lacks ablation on discretization choices. The trajectory discretization process (e.g.spatial quantization granularity, or temporal segmentation interval) may significantly influence expressiveness and planner performance. The paper does not provide ablations or analysis on these design factors.

**Questions:**

Same as weaknesses.

---

> ### Author Response · Authors · 2025-11-22
> **Response to Reviewer 5bV5**
>
> We really appreciate the reviewer for the constructive comments and positive feedback on our paper. Regarding the concerns of the reviewer 5bV5, we provide the following responses.
>
> > **W1**: Limited analysis of world model reliability when ego deviates from expert behavior. The frozen world model is assumed to remain accurate when the ego policy explores beyond regions well-covered by expert data. However, there is no case study or quantitative analysis showing how the world model behaves under large ego deviations or unusual interaction patterns, which could introduce compounding errors during RL fine-tuning.
>
> We appreciate the reviewer’s constructive comments. We examine the robustness of the frozen world model under ego-distribution shift by injecting zero-mean Gaussian perturbations of increasing magnitude ($\sigma = 0, 0.1, 0.3, 1, 3, 10 m$) into the ego state and measuring the resulting prediction errors for surrounding agents:
>
> |Noise $\sigma$ (m) |ADE (m) $\downarrow$|FDE (m) $\downarrow$|
> |-|-|-|
> |0|1.03|3.01|
> |0.1|1.07|3.15|
> |0.3|1.14|3.35|
> |1.0|1.22|3.53|
> |3.0|1.25|3.59|
> |10.0|1.26|3.59|
>
> As shown, the degradation remains small for realistic ego deviations ($\sigma < 1 m$), where ADE increases only 0.11 m and FDE increases 0.34 m. For larger perturbations ($\sigma ≥ 1 m$), the errors quickly saturate rather than diverge, and even with an unrealistically large perturbation of $\sigma = 10 m$, the FDE increases by only +0.58 m, showing no sign of exploding or compounding drift. **These results demonstrate that the frozen world model remains stable under ego perturbations.**
>
> This stability is expected because the world model conditions on each agent’s own motion history, map topology, and interactions with neighboring vehicles; the ego state is only one of many contextual signals. Therefore, moderate ego deviations do not dominate the multi-agent predictions, and even extreme deviations are anchored by the surrounding agents’ historical behaviors and the map structure, preventing unstable predictions.
>
>
> > **W2**: Tokenization design lacks ablation on discretization choices. The trajectory discretization process (e.g.spatial quantization granularity, or temporal segmentation interval) may significantly influence expressiveness and planner performance. The paper does not provide ablations or analysis on these design factors.
>
> We appreciate the reviewer's constructive comments. As described in Sec. 3.2 (lines 281-223), our tokenization design closely follows the K-disk motion discretization algorithm introduced in Trajeglish [1], while the spatial and temporal granularity (0.5 s step, 1024 motion tokens) is chosen to match the standard configuration used in SMART [2].   **Since tokenization is not the methodological focus of our work, we keep these settings consistent across all baselines to ensure a fair comparison.**
>
> Nevertheless, motivated by the reviewer’s suggestion, we conducted two additional ablations to test the robustness of VD-GRPO under different discretization settings. Our default configuration uses a 0.5 s interval with 1024 tokens. We further tested (A) 0.5 s with 512 tokens and (B) 1.0 s with 1024 tokens. Results are shown below:
>
> |Planner| 0.5s / 1024 tokens (default) | 0.5s / 512 tokens | 1.0s / 1024 tokens |
> |-|-|-|-|
> |Pre-training only|85.61|85.01|85.06|
> |+GRPO|88.65|86.37|85.85|
> |+VD-GRPO (Ours)|91.23|90.38|88.65|
>
> As expected, coarser spatial or temporal discretization slightly reduces pre-training accuracy due to lower expressiveness. Crucially, however, VD-GRPO consistently provides similar relative improvements over standard GRPO across all discretization choices. **This shows that the benefits of VD-GRPO are orthogonal to tokenization granularity**, and our conclusions on the effectiveness of VD-GRPO hold robustly regardless of discretization design.
>
>
> ___
> **References**
> [1] Jonah Philion, Xue Bin Peng, and Sanja Fidler. Trajeglish: Learning the language of driving scenarios. arXiv preprint arXiv:2312.04535, 2023.
> [2] Wei Wu, Xiaoxin Feng, Ziyan Gao, and Yuheng Kan. Smart: scalable multi-agent real-time motion generation via next-token prediction. Advances in Neural Information Processing Systems (NeurIPS), 2024.

---

> > ### Comment · Reviewer_5bV5 · 2025-11-24
> > **Official Comment by Reviewer 5bV5**
> >
> > Thank you for the author's patient reply! I believe this will contribute to the community.
> >
> > Best regards,
> > Reviewer 5bV5

---

> > > ### Author Response · Authors · 2025-11-25
> > >
> > > Thank you for your kind comment! We will release the code to further benefit the community. Thanks again for your thoughtful review.

---

### Official Review · Reviewer_eJWp · 2025-10-30

**Soundness:** 4
**Presentation:** 4
**Contribution:** 4
**Rating:** 8
**Confidence:** 4

**Summary:**

This paper presents Plan-R1, a two-stage framework that formulates trajectory planning as language modeling. The idea of decoupling behavior learning from principle alignment, inspired by LLM training paradigms, is both novel and elegant. The proposed Variance-Decoupled GRPO (VD-GRPO) effectively addresses the limitation of standard GRPO by preserving safety-critical gradients, leading to substantial performance gains on the nuPlan benchmark. The experimental section is comprehensive, including detailed ablations and clear visualizations, which convincingly demonstrate the method’s safety and feasibility improvements.

However, the theoretical justification of VD-GRPO (especially the fixed scaling constant) could be further strengthened, and an additional evaluation on another dataset (e.g., CARLA) would help validate generalization. Overall, this is a well-written and technically sound paper with strong empirical results and a creative conceptual contribution.

**Strengths:**

The paper introduces a novel and well-motivated idea of framing trajectory planning as language modeling through the two-stage Plan-R1 framework. The decoupling of behavior learning and principle alignment is conceptually elegant and practically effective. The proposed VD-GRPO clearly addresses a key limitation in standard GRPO, preserving safety-critical gradients and improving rare-event optimization. Experiments on the nuPlan benchmark are extensive and convincing, with strong gains in both non-reactive and reactive settings. The writing is clear, the figures are informative, and the work demonstrates a high level of technical maturity.

**Weaknesses:**

While the empirical results are strong, the theoretical justification of VD-GRPO remains limited. The fixed scaling constant is treated as a hyperparameter without principled analysis of its effect on convergence or stability. The dual-model setting, with a frozen world model, may introduce distribution drift in long-term interactions. Moreover, evaluation is restricted to nuPlan; results on other benchmarks such as CARLA or Waymo would strengthen claims of generalization.

**Questions:**

- Have the authors analyzed the learned motion tokens to see if they correspond to interpretable motion primitives or semantic driving actions?

- How would Plan-R1 perform under partial observability or sensor noise conditions compared to diffusion-based planners?

---

> ### Author Response · Authors · 2025-11-22
> **Response to Reviewer eJWp 1/4**
>
> We really appreciate the reviewer for the constructive comments and positive feedback on our paper. Regarding the concerns of the reviewer eJWp, we provide the following responses.
>
> > **W1**: While the empirical results are strong, the theoretical justification of VD-GRPO remains limited. The fixed scaling constant is treated as a hyperparameter without principled analysis of its effect on convergence or stability.
>
> We appreciate the reviewer’s constructive comments. We provide a concise derivation of how removing variance normalization and introducing $c$ affect the training dynamics. A complete derivation is provided in Appendix E.
>
> (1) Why removing GRPO’s variance normalization is necessary.
>
> The fine-tuning objective (Eq. (4)) is
> $$
> L_{\mathrm{finetune}}
> = -\frac{1}{GF}
> \sum^G_{g=1}\sum^F_{t=1}
> [\frac{\pi_e}{\pi_{e_{\mathrm{old}}}} \hat{A}^g_t
> -\beta\mathrm{KL}(\pi_e\|\pi_{\mathrm{ref}})],
> \tag{I}
> $$
> where $\hat{A}^g_t=\sum_{\tau\ge t}\tilde{R}(y_\tau^g).$ Standard GRPO normalizes rewards within each group as
> $$
> \tilde{R}^{\mathrm{GRPO}}(y_t^g)=
> \frac{R(y_t^g)-\mu_R}{\sigma_R},
> \tag{II}
> $$
> which yields the gradient (ignoring the KL term for clarity since its gradient does not interact with normalization):
> $$
> \nabla_\theta L_{\mathrm{GRPO}} =
> -\frac{1}{GF}
> \sum_{g,t}
> (\frac{1}{\sigma_R}
> \sum_{\tau\ge t}(R(y_{\tau}^g)-\mu_R))
> \nabla_\theta\log\pi_e.
> \tag{III}
> $$
> Thus, each group is effectively weighted by $1/\sigma_R$.
>
> Under our reward design (Eq. (3)), unsafe groups contain soft costs mixed with hard-zero penalties from violating safety constraints, whereas safe groups contain only soft-cost terms. This necessarily increases their dispersion, yielding
> $$
> \sigma_R(\text{unsafe}) > \sigma_R(\text{safe}).
> \tag{IV}
> $$
>
> Therefore, **GRPO systematically down-weights unsafe groups, even though these groups correspond to high-priority safety failures and should dominate the gradient.** VD-GRPO addresses this by replacing the variance $\sigma_R$ with a fixed constant $c$:
> $$
> \tilde{R}^{\mathrm{VD}}(y_t^g)=
> \frac{R(y_t^g)-\mu_R}{c},
> \tag{V}
> $$
> and its gradient becomes
> $$
> \nabla_\theta L_{\mathrm{VD-GRPO}} =
> -\frac{1}{c} \cdot \frac{1}{GF}
> \sum_{g,t}
> (
> \sum_{\tau\ge t}(R(y_{\tau}^g)-\mu_R))
> \nabla_\theta\log\pi_e.
> \tag{VI}
> $$
> Since the same constant $c$ applies to all groups, **VD-GRPO eliminates variance-induced reweighting and ensures that gradient magnitudes depend solely on true reward differences.**  Safe groups, which typically differ only in minor progress/comfort fluctuations, naturally contribute smaller gradients, while unsafe groups retain larger gradients reflecting their higher priority.
>
> (2) Why introducing the scaling constant $c$ is theoretically sound and practically necessary.
>
> (2.1) Theoretical soundness.
> From the VD-GRPO gradient expression (Eq. (VI)), the constant factor $1/c>0$ is a global scalar applied uniformly to all advantages. Such a scalar does not change the update direction, any stationary point, or the optimal policy; it only rescales the step size.
>
> (2.2) Practical necessity.
> Under our reward design (Eq. (3)), all rewards lie in $[0,1]$. As shown in Appendix E, the per-group reward standard deviation satisfies
> $$
> 0<\sigma_R \le 1/2.
> \tag{VII}
> $$
> Thus, GRPO implicitly amplifies advantages by a factor
> $$
> 1/\sigma_R \geq 2.
> \tag{VIII}
> $$
>
> The KL coefficient $\beta$ in Eq. (I) is tuned under this implicit amplification. If we simply remove $\sigma_R$ without compensation, the RL gradient becomes much smaller relative to the KL term, unintentionally causing the KL penalty to dominate and hindering effective policy improvement.
>
> To avoid retuning $\beta$ or the learning rate and to ensure a fair comparison with GRPO, we introduce a fixed constant $c \le 1$ to restore the RL term to the magnitude GRPO implicitly expects, while eliminating the variance-induced bias. As shown in Fig. 9, results remain stable over a wide range of $c$, with $c=0.1$ giving the best empirical performance; we therefore adopt it as the default.

---

> > ### Author Response · Authors · 2025-11-22
> > **Response to Reviewer eJWp 2/4**
> >
> > > **W2**: The dual-model setting, with a frozen world model, may introduce distribution drift in long-term interactions.
> >
> > We appreciate the reviewer’s constructive comments. We address it from two complementary perspectives:
> >
> > (1) Robustness of the frozen world model to planner-induced distribution shift. We examine the robustness of the frozen world model under ego-distribution shift by injecting zero-mean Gaussian perturbations of increasing magnitude ($\sigma = 0, 0.1, 0.3, 1, 3, 10 m$) into the ego state and measuring the resulting prediction errors for surrounding agents:
> > |Noise $\sigma$ (m) |ADE (m) $\downarrow$|FDE (m) $\downarrow$|
> > |-|-|-|
> > |0|1.03|3.01|
> > |0.1|1.07|3.15|
> > |0.3|1.14|3.35|
> > |1.0|1.22|3.53|
> > |3.0|1.25|3.59|
> > |10.0|1.26|3.59|
> >
> > As shown, the degradation remains small for realistic ego deviations ($\sigma < 1 m$), where ADE increases only 0.11 m and FDE increases 0.34 m. For larger perturbations ($\sigma ≥ 1 m$), the errors quickly saturate rather than diverge, and even with an unrealistically large perturbation of $\sigma = 10 m$, the FDE increases by only +0.58 m, showing no sign of exploding or compounding drift. **These results demonstrate that the frozen world model remains stable under ego perturbations.**
> >
> > This stability is expected because the world model conditions on each agent’s own motion history, map topology, and interactions with neighboring vehicles; the ego state is only one of many contextual signals. Therefore, moderate ego deviations do not dominate the multi-agent predictions, and even extreme deviations are anchored by the surrounding agents’ historical behaviors and the map structure, preventing unstable predictions.
> >
> > (2) How we mitigate the effect of world-model distribution drift on the planner.
> > - Short rollout horizon. During RL, the frozen world model is used only for short 8-s rollouts (nuPlan’s horizon), not long autoregressive simulations. As the robustness results above show, drift over this range remains small and bounded.
> > - Deterministic decoding for surrounding agents. We use top-1 decoding for surrounding-agent motion tokens, avoiding stochastic sampling errors that could amplify distribution drift.
> > - Mild world-model deviations can improve robustness. As noted in the classic *World Models* [1] paper, a learned environment that is not perfectly accurate can expose the planner to challenging or unseen scenarios, improving robustness. We agree with this view: slight deviations in the frozen world model act as implicit scenario augmentation, allowing the planner to train on corner-case interactions.
> >
> > Together, these properties ensure that distribution drift in the dual-model setup does not destabilize RL and can even improve planner robustness in practice.

---

> > > ### Author Response · Authors · 2025-11-22
> > > **Response to Reviewer eJWp 3/4**
> > >
> > > > **W3**: Evaluation is restricted to nuPlan; results on other benchmarks such as CARLA or Waymo would strengthen claims of generalization.
> > >
> > > We appreciate the reviewer's constructive comments. **Our experiments focus on nuPlan because it is currently the only public benchmark that offers both closed-loop multi-agent simulation and planning-level evaluation metrics, which are essential for evaluating a trajectory planner.** Other datasets unfortunately do not meet these requirements:
> > > - Argoverse / Argoverse 2: only open-loop prediction metrics with no closed-loop evaluators; prior work [1] shows that open-loop accuracy correlates weakly with planning quality.
> > > - Waymo Open Motion Dataset: supports motion prediction and sim agents, but lacks planning-level evaluators.
> > > - CARLA / nuScenes / NAVSIM: primarily used for end-to-end driving, requiring perception and control stacks and not directly compatible with planning models.
> > >
> > > Consequently, nearly all recent planning works, such as PlanTF [3], CarPlanner [4], GenDrive [5], and PLUTO [6], report results only on nuPlan. Diffusion Planner [7] additionally reports results on a private dataset. **Within this standard evaluation setting, we already perform extensive experiments across three representative nuPlan splits (Val14, Test14-random, Test14-hard) and under both reactive and non-reactive closed-loop configurations, consistently achieving state-of-the-art performance.**
> > >
> > > To further address the reviewer’s concern of generalization, we additionally evaluate Plan-R1 on the interPlan [8] benchmark, which has recently been adopted by some planning papers [9,10] as an auxiliary testbed for robustness. InterPlan is derived from nuPlan but systematically perturbs scenes (adding/removing agents, injecting obstacles, altering goals) to create challenging long-tail and interaction-heavy scenarios. The results are as follows (* indicates with rule-based post-processing; **bold** denotes best, *italic* denotes second best):
> > > |Type|Planner|Score|
> > > |-|-|-|
> > > |Expert|log-play|14.76|
> > > |Learning-based|UrbanDriver|5.56|
> > > ||PDM-Open|26.22|
> > > ||PlanTF|47.72|
> > > ||Diffusion Planner|50.07|
> > > ||PLUTO| **57.74**|
> > > ||Plan-R1 (Ours)|*56.64*|
> > > |Rule-based & Hybrid| IDM*|47.07|
> > > ||PLUTO*|63.88 |
> > > ||PDM-Closed*|*69.64*|
> > > ||Plan-R1* (Ours)|**72.33**|
> > > - Without post-processing, Plan-R1 achieves 56.64, slightly below PLUTO (57.74) but outperforming all other baselines by a clear margin. This is expected because interPlan constructs scenes by adding or removing agents and obstacles—an augmentation strategy closely aligned with PLUTO’s contrastive learning pipeline, naturally favoring it.
> > > - With post-processing, Plan-R1* achieves the best overall performance (72.33), outperforming PDM-Closed* by +2.69, demonstrating strong robustness in perturbed and long-tail scenarios.

---

> > > > ### Author Response · Authors · 2025-11-22
> > > > **Response to Reviewer eJWp 4/4**
> > > >
> > > > > **Q1**: Have the authors analyzed the learned motion tokens to see if they correspond to interpretable motion primitives or semantic driving actions?
> > > >
> > > > Thank you for the question. **Our motion tokens are not learned semantic driving actions.** As described in Sec. 3.2 (p.5, lines 218-223), they are generated offline using the K-disk clustering algorithm from Trajeglish [11], which partitions short-horizon motion increments into a fixed vocabulary of kinematic primitives. Each token directly corresponds to a (Δx,Δy,Δyaw) cluster center. Thus, the tokens are designed to represent quantized kinematic patterns, not high-level semantic actions (e.g., yielding, merging, overtaking). The model only learns embeddings for these pre-defined primitives; it does not discover semantic actions from data.
> > > >
> > > >
> > > > > **Q2**: How would Plan-R1 perform under partial observability or sensor noise conditions compared to diffusion-based planners?
> > > >
> > > > Thank you for the insightful question. We compared Plan-R1 with a representative diffusion-based planner (Diffusion Planner [7]) under two challenging settings:
> > > > - Partial observability: randomly dropping 10% of surrounding agents at each step
> > > > - Sensor noise: adding Gaussian noise to surrounding agents
> > > >     - position: $\mathcal{N}(0,0.1^2)$ m
> > > >     - heading: $\mathcal{N}(0,1^\circ{}^2)$
> > > >
> > > > |Setting|Diffusion Planner|Plan-R1 (Ours)|
> > > > |-|-|-|
> > > > |None (default)|89.19|91.23|
> > > > |Partial observability|85.75 (-3.44)|89.19 (-2.04)|
> > > > |Sensor noise|80.34 (-8.85)|89.83 (-1.40)|
> > > >
> > > > As shown, Plan-R1 not only achieves higher absolute performance, but also exhibits substantially smaller degradation (−2.04 vs −3.44; −1.40 vs −8.85) under both types of distribution shift. We attribute this robustness to (i) the inherent noise-tolerance of discrete autoregressive motion tokens, and (ii) the improved safety awareness introduced by rule-based RL fine-tuning.
> > > >
> > > >
> > > > **References**
> > > > [1] Ha, David, and Jürgen Schmidhuber. World models. arXiv preprint arXiv:1803.10122, 2018.
> > > > [2] Daniel Dauner, Marcel Hallgarten, Andreas Geiger, and Kashyap Chitta. Parting with misconceptions about learning-based vehicle motion planning. In Conference on Robot Learning (CoRL),2023.
> > > > [3] Jie Cheng, Yingbing Chen, Xiaodong Mei, Bowen Yang, Bo Li, and Ming Liu. Rethinking imitationbased planners for autonomous driving. In IEEE International Conference on Robotics and Automation (ICRA), 2024.
> > > > [4] Dongkun Zhang, Jiaming Liang, Ke Guo, Sha Lu, Qi Wang, Rong Xiong, Zhenwei Miao, and Yue Wang. Carplanner: Consistent auto-regressive trajectory planning for large-scale reinforcement learning in autonomous driving. arXiv preprint arXiv:2502.19908, 2025.
> > > > [5] Huang, Zhiyu, Xinshuo Weng, Maximilian Igl, Yuxiao Chen, Yulong Cao, Boris Ivanovic, Marco Pavone, and Chen Lv. Gen-drive: Enhancing diffusion generative driving policies with reward modeling and reinforcement learning fine-tuning. In IEEE International Conference on Robotics and Automation (ICRA), 2025.
> > > > [6] Jie Cheng, Yingbing Chen, and Qifeng Chen. Pluto: Pushing the limit of imitation learning-based planning for autonomous driving. arXiv preprint arXiv:2404.14327, 2024.
> > > > [7] Yinan Zheng, Ruiming Liang, Kexin Zheng, Jinliang Zheng, Liyuan Mao, Jianxiong Li, Weihao Gu, Rui Ai, Shengbo Eben Li, Xianyuan Zhan, et al. Diffusion-based planning for autonomous driving with flexible guidance. arXiv preprint arXiv:2501.15564, 2025.
> > > > [8] Hallgarten, Marcel, Julian Zapata, Martin Stoll, Katrin Renz, and Andreas Zell. Can vehicle motion planning generalize to realistic long-tail scenarios?. In IEEE/RSJ International Conference on Intelligent Robots and Systems (IROS), 2024.
> > > > [9] Sun, Qiao, Huimin Wang, Jiahao Zhan, Fan Nie, Xin Wen, Leimeng Xu, Kun Zhan, Peng Jia, Xianpeng Lang, and Hang Zhao. Generalizing motion planners with mixture of experts for autonomous driving. In IEEE International Conference on Robotics and Automation (ICRA), 2025.
> > > > [10] Wang, Lingguang, Ömer Şahin Taş, Marlon Steiner, and Christoph Stiller. FlowDrive: moderated flow matching with data balancing for trajectory planning. arXiv preprint arXiv:2509.21961, 2025.
> > > > [11] Jonah Philion, Xue Bin Peng, and Sanja Fidler. Trajeglish: Learning the language of driving scenarios. arXiv preprint arXiv:2312.04535, 2023.

---

> > > > > ### Author Response · Authors · 2025-11-27
> > > > > **Follow-up on Discussion**
> > > > >
> > > > > Dear reviewer eJWp,
> > > > >
> > > > > As the discussion period is approaching its end, **we wanted to kindly check whether there are any remaining questions or points that would benefit from further clarification.** We really appreciate the time and effort you have devoted to the review and discussion.
> > > > >
> > > > > Thank you very much and look forward to your replies!
> > > > >
> > > > > Best regards,
> > > > > Authors of Paper 12315

---

### Official Review · Reviewer_nc4d · 2025-10-30

**Soundness:** 2
**Presentation:** 2
**Contribution:** 2
**Rating:** 4
**Confidence:** 2

**Summary:**

The paper introduces a two-stage framework called Plan-R1 for safe, feasible trajectory planning in autonomous driving. In the first stage, the model learns diverse, human-like driving behaviors via pretraining on expert data. In the second stage, the model is fine-tuned with rule-based reinforcement learning (RL) to align trajectory planning with explicit principles such as safety, comfort, and traffic rules. To address the limitation of standard GRPO (Group Relative Policy Optimization)—which can dilute safety-critical signals in multi-objective planning—Plan-R1 proposes Variance-Decoupled GRPO (VD-GRPO), which preserves absolute reward magnitudes to ensure the dominance of safety objectives. Experiments show that Plan-R1 significantly improves planning safety and feasibility on the nuPlan benchmark, especially in challenging reactive settings.

**Strengths:**

• Clear two-stage framework: Plan-R1 decouples behavior learning from principle alignment, retaining human-like behavior while enhancing safety awareness and removing undesirable patterns present in expert data (p.1, lines 054–058).
• Novel VD-GRPO: In response to standard GRPO’s limitations, VD-GRPO replaces in-group normalization with centering and fixed scaling, effectively preventing rare but critical safety-violation signals from being washed out and ensuring safety-critical objectives dominate training (p.2, lines 075–083; p.6, lines 295–311).
• Rule-based rewards: Instead of relying on human preference data, Plan-R1 uses rule rewards that offer consistent, unbiased supervision, avoiding bias and improving scalability and reliability (p.3, lines 122–127; p.5, lines 266–269).
• Dual-model design: During RL fine-tuning, a trainable ego planner explores alternative decisions, while a frozen copy of the pretrained model acts as a reactive world model to predict other agents’ responses, enabling stable, interaction-aware joint prediction (p.2, lines 066–069; p.5, lines 256–265).
• Significant performance gains: On the nuPlan benchmark, Plan-R1 achieves state-of-the-art performance in both non-reactive and reactive settings, notably surpassing Diffusion Planner by +4.89, +7.98, and +7.11 points in the reactive setting, substantially improving safety and feasibility (p.7, Table 1, lines 370–377).
• Clear qualitative results: Figures 2 and 3 clearly show how Plan-R1 avoids undesirable behaviors common in pretrained or expert-data-only models, such as speeding, off-road driving, and collisions.

**Weaknesses:**

• Definition of pivots: The paper states that trajectories are discretized into motion tokens but does not delve into how these “pivot” points are chosen or defined, nor whether such discretization might miss key kinematic or geometric features (p.5, lines 217–223).
• Detailed analysis of VD-GRPO: Although VD-GRPO is proposed, there is limited theoretical analysis of how its parameters (e.g., the fixed scaling constant c) affect training dynamics; the discussion is mostly empirical (p.9, lines 453–460; p.15, lines 799–807).
• Missing user-uploaded images: The text mentions “Image generation: enabled,” but no actual images are provided, making it impossible to assess the quality or completeness of the figures referenced in the paper.

**Questions:**

1) In VD-GRPO, the choice of the fixed scaling constant (c) is empirical. Is there a more theoretical way to determine an optimal c, or is it highly task- and reward-design-dependent? (See p.15, lines 799–807.)
2) The paper notes that VD-GRPO “replaces in-group normalization with centering and fixed scaling.” Could you provide more concrete implementation details of this “fixed scaling,” and how it differs mathematically from traditional normalization methods? (See p.6, lines 302–303.)
3) In the ablation study of §4.3, Table 2 shows that VD-GRPO substantially improves the collision metric (+3.45) but has no effect on the comfort metric (+0.00). Could you explain in detail why VD-GRPO impacts some metrics more than others, and whether this relates to the high priority assigned to safety objectives in the reward function? (See p.8, Table 2; p.9, lines 453–458.)

---

> ### Author Response · Authors · 2025-11-22
> **Response to Reviewer nc4d 1/3**
>
> We thank the reviewer for the constructive comments. Regarding the concerns of the reviewer nc4d, we provide the following responses.
>
> > **W1**: Definition of pivots: The paper states that trajectories are discretized into motion tokens but does not delve into how these “pivot” points are chosen or defined, nor whether such discretization might miss key kinematic or geometric features (p.5, lines 217–223).
>
> **(1). Clarification on pivots.**  We guess there may be a misunderstanding, possibly due to confusion with other papers that use “pivot points.” **Our paper does not contain any notion of “pivots”.** The term does not appear anywhere in our submission. The relevant section (p.5, lines 217–223) only describes the construction of motion tokens, not pivot-based trajectory representations.
>
> **(2). Motion-token discretization and preservation of kinematic/geometric features.**  As stated on p.5 (lines 217–223), each motion token represents the displacement and heading change (Δx,Δy,Δyaw) over a fixed time step, following the standard Trajeglish [1] K-disk discretization. This retains the essential kinematic and geometric features of agents and is widely adopted in autoregressive trajectory models [1,2]. Importantly, all methods compared in our paper, including GRPO and VD-GRPO, use **the exact same motion-token vocabulary**, so the choice of discretization does not affect the fairness or validity of our comparative conclusions.

---

> > ### Author Response · Authors · 2025-11-22
> > **Response to Reviewer nc4d 2/3**
> >
> > > **W2**: Detailed analysis of VD-GRPO: Although VD-GRPO is proposed, there is limited theoretical analysis of how its parameters (e.g., the fixed scaling constant c) affect training dynamics; the discussion is mostly empirical (p.9, lines 453–460; p.15, lines 799–807).
> >
> > We appreciate the reviewer’s constructive comments. To address the concern regarding the theoretical role of the fixed scaling constant $c$, we provide a concise derivation of how removing variance normalization and introducing $c$ affect the training dynamics. A complete derivation is provided in Appendix E.
> >
> > (1) Why removing GRPO’s variance normalization is necessary.
> >
> > The fine-tuning objective (Eq. (4)) is
> > $$
> > L_{\mathrm{finetune}}
> > = -\frac{1}{GF}
> > \sum^G_{g=1}\sum^F_{t=1}
> > [\frac{\pi_e}{\pi_{e_{\mathrm{old}}}} \hat{A}^g_t
> > -\beta\mathrm{KL}(\pi_e\|\pi_{\mathrm{ref}})],
> > \tag{I}
> > $$
> > where $\hat{A}^g_t=\sum_{\tau\ge t}\tilde{R}(y_\tau^g).$ Standard GRPO normalizes rewards within each group as
> > $$
> > \tilde{R}^{\mathrm{GRPO}}(y_t^g)=
> > \frac{R(y_t^g)-\mu_R}{\sigma_R},
> > \tag{II}
> > $$
> > which yields the gradient (ignoring the KL term for clarity since its gradient does not interact with normalization):
> > $$
> > \nabla_\theta L_{\mathrm{GRPO}} =
> > -\frac{1}{GF}
> > \sum_{g,t}
> > (\frac{1}{\sigma_R}
> > \sum_{\tau\ge t}(R(y_{\tau}^g)-\mu_R))
> > \nabla_\theta\log\pi_e.
> > \tag{III}
> > $$
> > Thus, each group is effectively weighted by $1/\sigma_R$.
> >
> > Under our reward design (Eq. (3)), unsafe groups contain soft costs mixed with hard-zero penalties from violating safety constraints, whereas safe groups contain only soft-cost terms. This necessarily increases their dispersion, yielding
> > $$
> > \sigma_R(\text{unsafe}) > \sigma_R(\text{safe}).
> > \tag{IV}
> > $$
> >
> > Therefore, **GRPO systematically down-weights unsafe groups, even though these groups correspond to high-priority safety failures and should dominate the gradient.** VD-GRPO addresses this by replacing the variance $\sigma_R$ with a fixed constant $c$:
> > $$
> > \tilde{R}^{\mathrm{VD}}(y_t^g)=
> > \frac{R(y_t^g)-\mu_R}{c},
> > \tag{V}
> > $$
> > and its gradient becomes
> > $$
> > \nabla_\theta L_{\mathrm{VD-GRPO}} =
> > -\frac{1}{c} \cdot \frac{1}{GF}
> > \sum_{g,t}
> > (
> > \sum_{\tau\ge t}(R(y_{\tau}^g)-\mu_R))
> > \nabla_\theta\log\pi_e.
> > \tag{VI}
> > $$
> > Since the same constant $c$ applies to all groups, **VD-GRPO eliminates variance-induced reweighting and ensures that gradient magnitudes depend solely on true reward differences.**  Safe groups, which typically differ only in minor progress/comfort fluctuations, naturally contribute smaller gradients, while unsafe groups retain larger gradients reflecting their higher priority.
> >
> > (2) Why introducing the scaling constant $c$ is theoretically sound and practically necessary.
> >
> > (2.1) Theoretical soundness.
> > From the VD-GRPO gradient expression (Eq. (VI)), the constant factor $1/c>0$ is a global scalar applied uniformly to all advantages. Such a scalar does not change the update direction, any stationary point, or the optimal policy; it only rescales the step size.
> >
> > (2.2) Practical necessity.
> > Under our reward design (Eq. (3)), all rewards lie in $[0,1]$. As shown in Appendix E, the per-group reward standard deviation satisfies
> > $$
> > 0<\sigma_R \le 1/2.
> > \tag{VII}
> > $$
> > Thus, GRPO implicitly amplifies advantages by a factor
> > $$
> > 1/\sigma_R \geq 2.
> > \tag{VIII}
> > $$
> >
> > The KL coefficient $\beta$ in Eq. (I) is tuned under this implicit amplification. If we simply remove $\sigma_R$ without compensation, the RL gradient becomes much smaller relative to the KL term, unintentionally causing the KL penalty to dominate and hindering effective policy improvement.
> >
> > To avoid retuning $\beta$ or the learning rate and to ensure a fair comparison with GRPO, we introduce a fixed constant $c \le 1$ to restore the RL term to the magnitude GRPO implicitly expects, while eliminating the variance-induced bias. As shown in Fig. 9, results remain stable over a wide range of $c$, with $c=0.1$ giving the best empirical performance; we therefore adopt it as the default.
> >
> >
> > > **W3**: Missing user-uploaded images: The text mentions “Image generation: enabled,” but no actual images are provided, making it impossible to assess the quality or completeness of the figures referenced in the paper.
> >
> > **We guess there may alse be a misunderstanding, since our submission does not contain the phrase “Image generation: enabled” at any point.** We carefully re-checked the entire PDF and confirm that this phrase does not appear in any section. All figures referenced in the text (Fig. 1–9) are included and fully visible in the submitted manuscript. If the reviewer believes the phrase appears in a specific location, we would appreciate it being pointed out so we can address it precisely.

---

> > > ### Author Response · Authors · 2025-11-22
> > > **Response to Reviewer nc4d 3/3**
> > >
> > > > **Q1**: In VD-GRPO, the choice of the fixed scaling constant (c) is empirical. Is there a more theoretical way to determine an optimal c, or is it highly task- and reward-design-dependent? (See p.15, lines 799–807.)
> > >
> > > Thank you for the question. As shown in the concise theoretical derivation in the response to W2, **the constant $c$ plays the similar role as the KL coefficient $\beta$: it scales the RL term relative to the KL regularization but does not change the update direction, stationary points, or the optimal policy.** Because this balance depends on the reward scale (Eq. (3)), the task, and the chosen $\beta$, there is no closed-form “optimal” $c$. Its appropriate value is inherently task- and reward-design-dependent, just like choosing $\beta$ itself. Importantly, VD-GRPO is stable across a broad range of $c$ (Fig. 9), and $c=0.1$ provides the best result. Thus, selecting $c$ empirically, analogous to tuning $\beta$, is both standard and theoretically sufficient.
> > >
> > > > **Q2**: The paper notes that VD-GRPO “replaces in-group normalization with centering and fixed scaling.” Could you provide more concrete implementation details of this “fixed scaling,” and how it differs mathematically from traditional normalization methods? (See p.6, lines 302–303.)
> > >
> > > **Actually, the concrete implementation of “centering and fixed scaling” is already given explicitly on p.6 (Eq. (5)), where all mathematical components are fully specified.** For convenience, we restate the formulation and further highlight the difference from normalization used in standard GRPO.
> > >
> > > (1) Standard GRPO with in-group normalization. For each sampled trajectory group, standard GRPO normalizes the reward $R(y_t)$ as:
> > > $$\tilde{R}(y_t) = \frac{R(y_t) - \mu_R}{\sigma_R},$$
> > > where $\mu_R$ and $\sigma_R$ denote the mean and standard deviation of rewards computed within each group. This variance-based scaling erases cross-group magnitude differences, causing rare, high-variance safety-violation groups to have similar advantages as abundant low-variance safe groups, thereby suppressing optimization for safety-critical objectives.
> > >
> > > (2) Our VD-GRPO with centering and fixed scaling. VD-GRPO retains centering but replaces the variance-based scaling with a fixed constant $c$:
> > > $$\tilde{R}^{\mathrm{VD}}(y_t) = \frac{R(y_t) - \mu_R}{c}.$$
> > > Here, $c$ is a deterministic hyperparameter shared across groups. By avoiding variance-based scaling, VD-GRPO preserves the absolute magnitude gap between high-variance safety-violation groups and low-variance safe groups. As a result, safety-violation groups produce stronger advantages, exactly as illustrated in Figure 4 of our submission, allowing safety objectives to remain dominant throughout training.
> > >
> > > > **Q3**: In the ablation study of §4.3, Table 2 shows that VD-GRPO substantially improves the collision metric (+3.45) but has no effect on the comfort metric (+0.00). Could you explain in detail why VD-GRPO impacts some metrics more than others, and whether this relates to the high priority assigned to safety objectives in the reward function? (See p.8, Table 2; p.9, lines 453–458.)
> > >
> > > Thank you for your careful observation. It is true that VD-GRPO improves some metrics (e.g., collision) more than others (e.g., comfort), and this behavior is reasonable for the following two reasons:
> > > - Different metrics have different headroom for improvement. As shown in Table 2, the pretrained model already achieves near-saturated comfort performance (99.62), leaving almost no room for further gains. In contrast, its collision score is lower (94.83), leaving substantial headroom. VD-GRPO therefore improves metrics where the pretrained model is deficient, while comfort remains unchanged simply because it is already near-optimal (99.62).
> > > - VD-GRPO preserves reward-scale priorities, emphasizing safety over comfort. Our reward function assigns much higher priority to safety than to comfort (p.6, lines 270–278). This follows the standard and widely accepted principle in autonomous driving: safety must always dominate comfort [3]. By removing GRPO’s variance normalization and preserving absolute reward scales, VD-GRPO naturally allocates more optimization capacity to high-priority safety signals. This leads to strong improvements on collision, while comfort—being low-weight and already saturated—changes little.
> > >
> > > **References**
> > > [1] Jonah Philion, Xue Bin Peng, and Sanja Fidler. Trajeglish: Learning the language of driving scenarios. arXiv preprint arXiv:2312.04535, 2023.
> > > [2] Wei Wu, Xiaoxin Feng, Ziyan Gao, and Yuheng Kan. Smart: scalable multi-agent real-time motion generation via next-token prediction. Advances in Neural Information Processing Systems (NeurIPS), 2024.
> > > [3] Caesar, Holger, Juraj Kabzan, Kok Seang Tan, Whye Kit Fong, Eric Wolff, Alex Lang, Luke Fletcher, Oscar Beijbom, and Sammy Omari. nuplan: A closed-loop ml-based planning benchmark for autonomous vehicles. arXiv preprint arXiv:2106.11810, 2021.

---

> > > > ### Author Response · Authors · 2025-11-27
> > > > **Follow-up on Discussion**
> > > >
> > > > Dear reviewer nc4d,
> > > >
> > > > As the discussion period is approaching its end, **we wanted to kindly check whether there are any remaining questions or points that would benefit from further clarification.** We really appreciate the time and effort you have devoted to the review and discussion.
> > > >
> > > > Thank you very much and look forward to your replies!
> > > >
> > > > Best regards,
> > > > Authors of Paper 12315

---

### Author Response · Authors · 2025-12-02
**Summary**

Dear ACs, SACs, PCs, and Reviewers,

Thank you for your dedication to this conference. As the rebuttal phase comes to a close, we would like to summarize key points from the reviews and express our sincere gratitude to all participants.

Our proposed VD-GRPO was recognized as a novel and principled improvement over standard GRPO, effectively preserving safety-critical gradients (Reviewers nc4d, eJWp, 5bV5, fE6E). All reviewers emphasized the SOTA performance of Plan-R1 on nuPlan, noting substantial safety gains supported by strong ablations (Reviewers nc4d, eJWp, 5bV5, fE6E). The dual-model rollout was praised as an effective and stable mechanism for interaction-aware simulation during RL fine-tuning (Reviewers nc4d, 5bV5, fE6E).  The decoupling of behavior learning and principle alignment of our two-stage Plan-R1 framework was regarded as clear, elegant and practically effective (Reviewers nc4d, eJWp).

Alongside these positive highlights, reviewers also provided constructive and insightful suggestions. In response, we conducted additional experiments and analyses, including a theoretical justification of VD-GRPO, an evaluation of the frozen world model’s robustness, and a comprehensive assessment on interPlan. These results have been incorporated into the appendix. We sincerely thank the reviewers for these valuable comments, which helped us further strengthen the work.


Prior to the platform incident, the reviewers' attitudes had become increasingly positive, reflecting growing endorsement of the paper's value:
- Reviewer 5bV5 commented (24 Nov): "Thank you for the author's patient reply! I believe this will contribute to the community."
- Reviewer fE6E commented (23 Nov): "I would like to thank the authors for their comprehensive and detailed rebuttal. More experiments and analysis on interPlan have shown the effectiveness of their proposed methods, and the ablation study with respect to world model and planner clearly shows the intuition and effectiveness of the authors' designs. Based on these, my concerns have been resolved, and I would like to increase my score from 6 to 8."

The remaining two reviewers nc4d and eJWp had not yet responded before the platform incident. Their concerns were primarily requests for a theoretical justification of VD-GRPO, clarifications on implementation details such as motion-token construction, and questions that overlapped with those raised by Reviewers 5bV5 and fE6E, including the robustness of the frozen world model and the need for evaluation beyond nuPlan. We provided detailed theoretical analysis, comprehensive clarifications, and new experiments addressing these points. Reviewers 5bV5 and fE6E, who raised the same concerns, explicitly stated that these additions fully resolved their issues. Given this alignment, we believe the concerns from reviewers nc4d and eJWp have likewise been satisfactorily addressed.

Finally, we express our sincere gratitude once again to all reviewers and chairs. Your thoughtful feedback has helped us refine, strengthen, and broaden the impact of this work. We hope you will appreciate our efforts and support its broader visibility.

Sincerely,

Authors of Paper 12315

---

### Meta-Review · Area_Chair_JcZW · 2025-12-16

**Summary:**

This submission receives 4, 6, 8, 8 scores. The reviewers' initial concerns focused primarily on the technical depth of the proposed VD-GRPO method, the robustness of the dual-model design, and the scope of the experimental validation. The authors provided comprehensive rebuttals. One reviewer with 6 (marginally above) explicitly said he or she would increase it to 8.

AC has gone through the paper, the reviews, and the discussion. In particular, the reviewer with a 4 score raised the issue of pivot, but the term does not appear anywhere in the submission. Also “Image generation: enabled,”, but no such things in the submission. It is likely the reviewer utilized GenAI to generate reviews. That undermines the credibility of the reviewer. The authors also provided a detailed analysis and derivation of VD-GRPO. Thus, AC sides with the other three very positive reviewers and recommended acceptance.

**Reviewer Concerns:**

Please see the summary above. Most of the concerns have been addressed. Unfounded concerns from the bad reviewer are not addressable.

**Reviewer Scores:**

6,8,8,8

---

### Decision · Program_Chairs · 2026-01-26

Accept (Poster)